# Octic Vision Transformers: Quicker ViTs Through Equivariance

## Abstract

Why are state-of-the-art Vision Transformers (ViTs) not designed to exploit natural geometric symmetries such as 90-degree rotations and reflections? In this paper, we argue that there is no fundamental reason, and what has been missing is an efficient implementation. To this end, we introduce Octic Vision Transformers (octic ViTs) which rely on octic group equivariance to capture these symmetries. In contrast to prior equivariant models that increase computational cost, our octic linear layers achieve 5.33x reductions in FLOPs and up to 8x reductions in memory compared to ordinary linear layers. In full octic ViT blocks the computational reductions approach the reductions in the linear layers with increased embedding dimension. We study two new families of ViTs, built from octic blocks, that are either fully octic equivariant or break equivariance in the last part of the network. Training octic ViTs supervised (DeiT-III) and unsupervised (DINOv2) on ImageNet-1K, we find that they match baseline accuracy while at the same time providing substantial efficiency gains.

## 1 Introduction

In the pursuit of flexible yet scalable models, Vision Transformers (ViTs) (Dosovitskiy et al., 2021) have emerged as the dominant architecture in modern computer vision. Key to their success is the combination of visual tokens, constructed from image patches, with the powerful attention mechanism (Vaswani et al., 2017), resulting in a versatile and scalable architecture. This scalability is due in large part to weight-sharing between the tokens, which ensures permutation equivariance.

Equivariance provides a powerful inductive bias in neural networks by enforcing structured responses to transformations such as permutations, translations, rotations or reflections. Another major benefit of equivariance is the potential for reducing computational costs, for instance by token-wise weight-sharing as mentioned. In this paper we obtain further computational reductions by parameterizing ViTs in the Fourier domain of a symmetry group. Concretely, we imbue ViTs with equivariance under roto-reflections, formalized through the octic group $D_8$. Our implementation is closely aligned with standard ViTs, but the equivariance under $D_8$ allows us to implement the token-wise linear layers in the Fourier domain, making them faster than standard linear layers.

$D_8$ equivariance was introduced to Convolutional Neural Networks (CNNs) by Cohen & Welling (2016), demonstrating improved parameter efficiency through weight sharing (Wood & Shawe-Taylor, 1996; Bekkers et al., 2018; Weiler & Cesa, 2019). Yet, despite its theoretical appeal, state-of-the-art vision models (Radford et al., 2021; Kirillov et al., 2023; Oquab et al., 2024; Wang et al., 2024a) do not incorporate group equivariance other than the permutation equivariance of transformer layers. We argue that this is not due to a lack of utility, but due to practical limitations: most existing implementations construct equivariant layers using computationally inefficient architectures, leading to increased FLOPs and runtime. Without efficient, hardware-compatible implementations, these methods have remained impractical for large-scale models. As a result, the potential of equivariant design remains largely unexplored in state-of-the-art systems, in particular in the context of ViTs. Recently, Bökman et al. (2025) showed that equipping ViTs with horizontal flip equivariance results in retained performance while saving FLOPs. However, their improvements are limited due to the small cardinality of their chosen group.

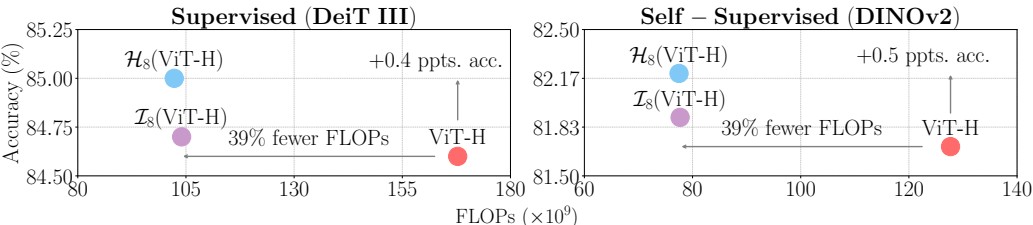

Figure 1: **Computational savings**. Using octic layers in ViTs significantly reduces the computational complexity without sacrificing accuracy on ImageNet-1K, for both supervised and self-supervised training. Detailed results can be found in Section 4.

In this paper, we demonstrate that scaling equivariance to larger groups can be efficiently implemented – yielding faster, stronger, and more compact models, cf. Figure 1. Specifically, we introduce octic-equivariant layers for ViTs, leveraging the $D_8$ symmetry group of $90°$ rotations and reflections. Our approach integrates seamlessly into existing ViT architectures and leads to significant gains in throughput and memory efficiency, without sacrificing accuracy.

In summary, our contributions are as follows:

(a) We introduce octic-equivariant layers for ViTs, described in Section 3. In our implementation, octic-equivariant linear layers use $5.33$ times fewer FLOPs and $8$ times less memory per feature dimension than ordinary linear layers. Octic-equivariant ViT layers hence asymptotically have the same compute savings (Table 1, Figure 4a).

(b) We propose two highly competitive families of ViTs, that are either fully rotation equivariant ($\mathcal{I}_8$) or break equivariance in late layers of the network ($\mathcal{H}_8$).

(c) In Section 4, we demonstrate empirically that our ViTs can be used in state-of-the-art ViT training recipes (DeiT III and DINOv2) without re-tuning hyperparameters. In particular, we achieve a $40\%$ FLOP saving with our $\mathcal{I}_8(\text{ViT-H})$ and $\mathcal{H}_8(\text{ViT-H})$ models while matching baseline performance (Figure 1).

(d) We study the effects of different methods of invariantization (testing six different methods in Appendix F), completely breaking equivariance (Section 4.3) and varying the number of equivariant layers (Section 4.3). These ablations will help guide future research on equivariant architectures at scale.

In addition to introducing new state-of-the-art ViTs, from a broader perspective our contributions give clear evidence that equivariance can matter at scale, which has been a subject of debate in recent literature (Abramson et al., 2024; Wang et al., 2024b; Brehmer et al., 2025; Bökman et al., 2025).

## 2 RELATED WORK

**Vision Transformers.** The ViT was introduced by Dosovitskiy et al. (2021) and has subsequently achieved state-of-the-art results in many domains of computer vision (Carion et al., 2020; Radford et al., 2021; Kirillov et al., 2023; Edstedt et al., 2024; Wang et al., 2025). Significant efforts have been made to scale ViTs (Zhai et al., 2022; Dehghani et al., 2023), alongside strategies to do so efficiently (Alabdulmohsin et al., 2023). In this work, we propose incorporating octic layers in ViTs, which not only maintain efficiency at larger scales but become increasingly effective as model size grows, thus directly leveraging the scaling of ViTs. Hierarchical Transformers (Liu et al., 2021; Wu et al., 2021; Hassani et al., 2023) use translational symmetry, and SparseViT (Chen et al., 2023) extends this with sparse activations. In contrast, our work instead focuses on roto-reflections and builds exact equivariance into the architecture.

**Equivariant Networks.** The equivariance of CNNs to (cyclic) image translations can be extended to incorporate larger symmetry groups such as rotations and reflections, as shown by Cohen & Welling (2016); Dieleman et al. (2016) using Group Equivariant CNNs (G-CNNs). Cohen & Welling

(2017); Weiler & Cesa (2019) generalized G-CNNs to steerable CNNs, where the features transform according to general group representations. Our octic ViTs can be seen as a more scalable ViT analogue of the octic steerable CNNs by Cohen & Welling (2017). There have also been prior efforts on attention- and Transformer-based equivariant architectures. For both point clouds (Fuchs et al., 2020; Hutchinson et al., 2021; Assaad et al., 2023; Liao & Smidt, 2023) and, more closely to ours, images (Romero et al., 2020; Xu et al., 2023; Rojas-Gomez et al., 2024; Kundu & Kondor, 2025). However, these prior works do not obtain computational benefits over non-equivariant transformers, in contrast to our ViTs. Our work is part of an ongoing research direction of studying and improving the scalability of equivariant networks (Bekkers et al., 2024; Brehmer et al., 2025; Bharadwaj et al., 2025; Vadgama et al., 2025; NVIDIA). Prior work in this direction mostly focused on point cloud data, with the notable exception of He et al. (2021) and Bökman et al. (2025) who considered images. He et al. (2021) increased the computational efficiency of G-CNNs but in contrast to our work did not achieve such benefits against standard networks. Directly inspiring our work, Bökman et al. (2025) demonstrated that incorporating horizontal mirroring equivariance into modern image classifiers increases compute efficiency while maintaining representational power. In contrast, we consider the larger octic group and thus achieve further savings in FLOPs. We conduct more extensive experimentation than Bökman et al. (2025) and address open questions in their work by studying the effect of breaking equivariance, invariantization and the number of equivariant blocks.

## 3 METHOD

In this section, we design octic-equivariant ViT layers. We begin with preliminaries for octic equivariance in Section 3.1, followed by the introduction of octic ViTs in Section 3.2, specifics of the Transformer layers in Section 3.3, and a detailed discussion of computational efficiency in Section 3.4. We summarize the most important notation in Section 3.1.1.

### 3.1 PRELIMINARIES ON OCTIC EQUIVARIANCE

In this work we focus on the dihedral group with eight elements, $D_8 = \{e, r, r^2, r^3, s, sr, sr^2, sr^3\}$, such that $r^4 = s^2 = e$ is the identity element and $r^3 = srs$.[1] We think of $D_8$ as acting on images by reflections $s$ and 90° rotations $r$. $D_8$ is also called the octic group, and we opt for this shorter name throughout.

We consider network layers as maps between real group representations of $D_8$. In our setting, a real representation is a vector space $\mathbb{R}^n$ equipped with a group homomorphism $\rho$ from $D_8$ to the group of $n \times n$ invertible real matrices. In other words for every $g \in D_8$, $\rho(g)$ is an invertible matrix and for every $g, h \in D_8$, $\rho(gh) = \rho(g)\rho(h)$. $\rho$ specifies how $x \in \mathbb{R}^n$ transforms under $D_8$. A representation of $D_8$ is also defined by choosing matrices $\rho(r)$ and $\rho(s)$ such that $\rho(r)^4 = \rho(s)^2 = I$ and $\rho(r)^{-1} = \rho(s)\rho(r)\rho(s)$. There are only a few different representations of $D_8$ used in this work, we list them below as examples. Finally, it is worth mentioning that all representations considered here are orthogonal, i.e., satisfy $\rho(g)^{-1} = \rho(g)^\mathsf{T}$.

The atomic building blocks of group representations are the so-called irreducible representations.

**Example 3.1** (Irreducible representations)**.** The five irreducible representations, short irreps, of $D_8$ are defined by

$$
\begin{aligned}
&\rho_{A1}(r) = \rho_{A1}(s) = 1; &&\rho_{A2}(r) = 1,\ \rho_{A2}(s) = -1;\\
&\rho_{B1}(r) = -1,\ \rho_{B1}(s) = 1; &&\rho_{B2}(r) = \rho_{B2}(s) = -1;\\
&\text{and}\quad \rho_E(r) = \begin{pmatrix} 0 & -1 \\ 1 & 0 \end{pmatrix},\ \rho_E(s) = \begin{pmatrix} -1 & 0 \\ 0 & 1 \end{pmatrix}.
\end{aligned}
\tag{1}
$$

We use the same notation as the original work on steerable CNNs (Cohen & Welling, 2017) for these irreps, but choose a different basis for $\rho_E$. It is known from elementary representation theory that any representation of $D_8$ can be decomposed into irreps as

$$
\rho(g) = Q \left( \bigoplus_{i \in \{A1, A2, B1, B2, E\}} m_i \rho_i(g) \right) Q^{-1}
\tag{2}
$$

where $\oplus$ denotes direct sum of representations, or stacking matrices in a block diagonal, and we write $m_i \rho_i(g)$ for $\oplus$'ing $\rho_i(g)$ with itself $m_i$ times. Here $Q$ is an invertible matrix and the $m_i$ are integers specifying the multiplicity of each irrep.

---

[1]The reader is cautioned that $D_8$ is sometimes alternatively denoted $D_4$.

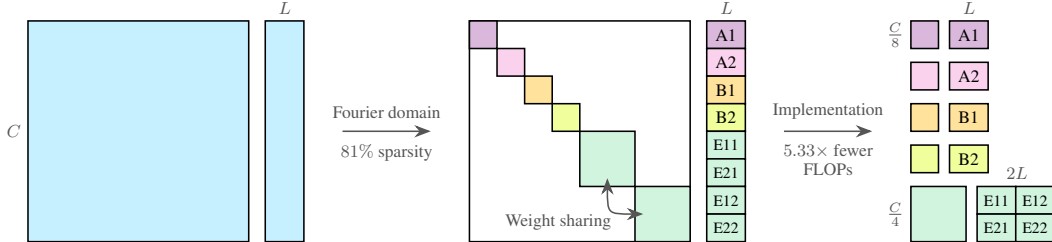

Figure 2: $D_8$ **Linear layers.** Implementing equivariant linear layers in the Fourier domain of $D_8$ gives a major computational benefit. **Left:** A $C \times C$ weight matrix being multiplied by $L$ tokens of feature dimension $C$. **Center:** The block-diagonalization that happens when enforcing the layer to be $D_8$-equivariant in the Fourier domain. More precisely, we enforce equivariance with respect to the representation $\frac{C}{8}\rho_{\text{iso}}$ that splits into irreps $\rho_{\text{A1}}, \rho_{\text{A2}}, \rho_{\text{B1}}, \rho_{\text{B2}}$ and $\rho_{\text{E}}$ as detailed in Section 3. There is no mixing between different irreps and the weight sharing in the block-diagonal stems from the fact that $\rho_{\text{E}}$ is a two-dimensional irrep. **Right:** An efficient implementation of the original $C \times C$ by $C \times L$ matrix multiplication as four $\frac{C}{8} \times \frac{C}{8}$ by $\frac{C}{8} \times L$ and one $\frac{C}{4} \times \frac{C}{4}$ by $\frac{C}{4} \times 2L$ matrix multiplication. An equivariant linear layer of this type requires $16/3 \approx 5.33$ times fewer FLOPs to compute and has 8 times fewer parameters than the ordinary linear layer shown to the left.

**Example 3.2** (Regular representation). The regular representation $\rho_{\text{reg}}$ can be thought of as $D_8$ acting canonically on the vector space of functions $\phi : D_8 \to \mathbb{R}$:

$$[\rho_{\text{reg}}(g)\phi](h) = \phi(g^{-1}h). \tag{3}$$

We identify each $\phi : D_8 \to \mathbb{R}$ with the vector

$$\begin{pmatrix} \phi(e) & \phi(r^3) & \phi(r^2) & \phi(r) & \phi(s) & \phi(sr^3) & \phi(sr^2) & \phi(sr) \end{pmatrix}^{\mathsf{T}} \in \mathbb{R}^8 \tag{4}$$

so that $\rho_{\text{reg}}(g)$ can be written as a permutation matrix. Importantly, $\rho_{\text{reg}}$ commutes with pointwise activation functions such as GELU (Hendrycks & Gimpel, 2016).

**Example 3.3** (Isotypical decomposition / Fourier transform). The regular representation $\rho_{\text{reg}}$ can be block-diagonalized to its isotypical decomposition $\rho_{\text{iso}}$ through equation 2 as $\rho_{\text{reg}}(g) = Q_{\text{reg}}\rho_{\text{iso}}(g)Q_{\text{reg}}^{-1}$ with

$$\rho_{\text{iso}}(g) = \rho_{\text{A1}}(g) \oplus \rho_{\text{A2}}(g) \oplus \rho_{\text{B1}}(g) \oplus \rho_{\text{B2}}(g) \oplus 2\rho_{\text{E}}(g). \tag{5}$$

It is a general fact for finite groups that the regular representation decomposes into all irreps with multiplicities equal to their dimensions. The change of basis $Q_{\text{reg}}$ (written out in full in Appendix A) is the inverse Fourier transform of $D_8$, with $Q_{\text{reg}}^{-1} = Q_{\text{reg}}^{\mathsf{T}}$ being the Fourier transform.

Equivariant networks use equivariant linear layers, which map between representations. A linear $G$-equivariant map or intertwiner, $W \in \mathbb{R}^{n \times n}$, between representations $\rho_1, \rho_2$, commutes with their action, i.e., $\rho_2(g)W = W\rho_1(g)$. It follows from Schur's lemma that $W = \lambda I$ for some scalar $\lambda$ if $\rho_1, \rho_2$ are irreps, and that $\lambda = 0$ if $\rho_1, \rho_2$ are not isomorphic (Serre, 1977, Section I.2.2)[2]. As $\rho_{\text{iso}}$ is just a stack of irreducible representations, any intertwiner between such representations will be sparse, which is why they are less computationally expensive than ordinary linear layers. The naive computational complexity for a linear map $W : \mathbb{R}^{|D_8|} \to \mathbb{R}^{|D_8|}$ is $|D_8|^2 = 64$ multiplications[3]. In contrast, intertwiners $\rho_{\text{iso}} \to \rho_{\text{iso}}$ require a total of $\sum_i^k m_i^2 d_i = 1 + 1 + 1 + 1 + 2^2 \cdot 2 = 12$ multiplications for $D_8$. From a signal processing perspective, this is analogous to the computational saving of convolution being point-wise multiplications in frequency space. We visualize the computational savings obtained by working in the Fourier domain of $D_8$ in Figure 2.

**Example 3.4** (Images). Square images can be considered as elements of $\mathbb{R}^{3 \times M \times M}$ where 3 is the number of color channels and $M$ is the image height/width in pixels. There is a natural $D_8$-representation $\rho_{\text{image}}$ associated with square images, where $\rho_{\text{image}}(r)$ is the pixel permutation rotating the images anti-clockwise by 90° and $\rho_{\text{image}}(s)$ is the permutation reflecting the images left-to-right.

---

[2]We can apply Schur's lemma for complex representations here since the irreps listed in Example 3.1 are irreducible over the complex numbers. We will however only use real-valued linear layers, i.e. $\lambda \in \mathbb{R}$.

[3]We ignore the additions for simplicity.

**Example 3.5** (ViT features). In ViTs, features are elements of $\mathbb{R}^{C \times N \times N}$, which we will think of as $C \times N^2$ matrices. Here, $N$ is the number of image tokens along the height/width of the image, so $N = M/P$ where $P$ is the patch size (typically $P = 14$ or $P = 16$) and $C$ is the channel dimension. The simplest representation that we consider on features is the permutation representation $\rho_{\text{token}}$ that, analogously to $\rho_{\text{image}}$, permutes the tokens according to elements of $D_8$.

**Example 3.6** (Steerable ViT features). We can equip the channel dimension of ViT features with a group representation $\rho_{\text{chan}}$ to obtain "steerable" features. If $C$ is divisible by 8, we can consider multiples of $\rho_{\text{reg}}$ or $\rho_{\text{iso}}$ as $\rho_{\text{chan}}$. The complete representation $\rho$ acting on the $C \times N^2$-matrix $\mathbf{x}$, is then permuting the tokens according to $\rho_{\text{token}}$ and modifying the channels according to $\rho_{\text{chan}}$. Concretely,

$$\rho(g)\text{Vec}(\mathbf{x}) = \text{Vec}\left(\rho_{\text{chan}}(g)\mathbf{x}\rho_{\text{token}}(g)^{\mathsf{T}}\right) = \left(\rho_{\text{token}}(g) \otimes \rho_{\text{chan}}(g)\right)\text{Vec}(\mathbf{x}), \tag{6}$$

where $\text{Vec}(\mathbf{x})$ is the column-wise vectorization of the matrix $\mathbf{x}$ and $\otimes$ is the tensor product of representations or (equivalently) the Kronecker product of matrices. We refer to features transforming according to equation 6 as $\rho_{\text{chan}}$-steerable, or features of type $\rho_{\text{chan}}$. This is a simpler form of the induced representations typically considered in steerable CNNs (Cohen & Welling, 2017; Weiler & Cesa, 2019), the simplification coming from the fact that we don't enforce translation equivariance. Steerable ViT-features are illustrated in the Appendix, Figure 5.

**Example 3.7** (Patchification of images). One can consider a patchified image as a steerable ViT feature in the following way. By patchification we mean the operation of reshaping a $3 \times M \times M$ image first to $N^2$ patches of size $P \times P$, with $NP = M$ and then to a $3P^2 \times N^2$ matrix. When we transform the original image by $\rho_{\text{image}}$, the patchified image is rotated by $\rho_{\text{token}}$ and $\rho_{\text{chan}}$ as in equation 6. Now $\rho_{\text{chan}}(g)$ is a permutation matrix that rotates or mirrors a patch and we will denote this particular $\rho_{\text{chan}}$ by $\rho_{\text{patch}}$.

### 3.1.1 NOTATION

For the convenience of the reader, we collect the most important notation in the paper in this section. We use the bold letter $\mathbf{x}$ for ViT features, which have shape $C \times L$, where $L$ is the number of tokens and $C$ the channel dimension. Typically, $L = N^2$, where $N$ is the image height/width in tokens, or $L = N^2 + 1$ with a class token. For an individual $C$-dimensional token we use the letter $x$ which is often acted on by $\rho_{\text{chan}} = \frac{C}{8}\rho_{\text{iso}}$, in which case we can split $x$ into $C/8$-dimensional sub-tokens $x_{\text{A1}}, x_{\text{A2}}, x_{\text{B1}}, x_{\text{B2}}, x_{\text{E11}}, x_{\text{E12}}, x_{\text{E21}}, x_{\text{E22}}$, where the first four transform according to the irreps $\rho_{\text{A1}}, \rho_{\text{A2}}, \rho_{\text{B1}}, \rho_{\text{B2}}$ respectively while the $2 \times \frac{C}{8}$-matrices $\begin{pmatrix} x_{\text{E11}} & x_{\text{E12}} \end{pmatrix}^{\mathsf{T}}$ and $\begin{pmatrix} x_{\text{E21}} & x_{\text{E22}} \end{pmatrix}^{\mathsf{T}}$ both transform according to $\rho_{\text{E}}$.

### 3.2 OCTIC ViTs

We construct ViT versions that map images to steerable ViT features $\mathbf{x} \in \mathbb{R}^{C \times L}$ in the first layer (PatchEmbed) and then process steerable ViT features in subsequent layers. We choose $\rho_{\text{chan}}$ to be a multiple of $\rho_{\text{iso}}$, enabling efficient linear layers. Typically, we write $\rho_{\text{chan}} = \frac{C}{8}\rho_{\text{iso}}$ where $C$ is the embedding dimension of the ViT. For classification tasks, we map the steerable ViT features to $D_8$ invariant features fed into a classification head. We denote ViTs that use $\rho_{\text{chan}}$-steerable features in all layers as $D_8(\text{ViT})$. We will also consider networks that use $\rho_{\text{chan}}$-steerable features for the first layers and then map either to $D\rho_{\text{A1}}$-steerable features, these networks are denoted $\mathcal{I}_8(\text{ViT})$, or break equivariance, denoted $\mathcal{H}_8(\text{ViT})$. We refer to ViTs that fall into these three families broadly as octic ViTs. In Section 4.3, we study the effect of the number of octic layers.

A commonly appreciated fact is that Transformer components $b$ such as MLP and Attention are permutation equivariant over tokens, which implies that

$$b(\mathbf{x}\rho_{\text{token}}(g)^{\mathsf{T}}) = b(\mathbf{x})\rho_{\text{token}}(g)^{\mathsf{T}}. \tag{7}$$

For $b$ to be fully equivariant it also needs to be equivariant in the channel dimension

$$b\left(\rho_{\text{chan}}(g)\mathbf{x}\rho_{\text{token}}(g)^{\mathsf{T}}\right) = \rho_{\text{chan}}(g)b(\mathbf{x})\rho_{\text{token}}(g)^{\mathsf{T}}. \tag{8}$$

Designing the components of octic ViT blocks so that they satisfy equation 8 is the topic of Section 3.3.

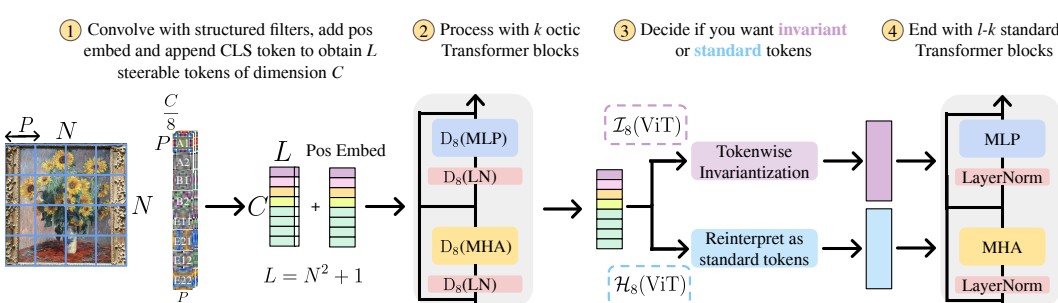

Figure 3: **Architecture**. Patches are first extracted from an image using specialized octic filters and the resulting features are processed by $k$ octic ViT blocks. The final embeddings can be fed to $l - k$ standard Transformer blocks (as demonstrated by our $\mathcal{H}_8$ and $\mathcal{I}_8$ ViTs). When $k = l$, we denote $\mathcal{I}_8(\text{ViT})$ by $D_8(\text{ViT})$, which hence only uses octic ViT blocks before a final invariantization.

### 3.3 OCTIC LAYERS

In this section we detail our implementations of $D_8$ equivariant Transformer layers. Together, these pieces can be combined into a ViT block $b = b_2 \circ b_1$ where

$$b_1(\mathbf{x}) = \mathbf{x} + \text{MHA}(\text{LN}(\mathbf{x})), \quad b_2(\mathbf{x}) = \mathbf{x} + \text{MLP}(\text{LN}(\mathbf{x})). \tag{9}$$

LN is layer normalization, MHA multi-head self-attention and MLP a $C \to 4C$ linear map, GELU, and $4C \to C$ linear map. The blocks can subsequently be stacked, as illustrated in Figure 3.

#### 3.3.1 THE PATCH EMBEDDING LAYER

The first layer in a ViT, following (Dosovitskiy et al., 2021), is the patch embedding, short PatchEmbed. In our case, it can be viewed as a mapping from steerable features of type $\rho_{\text{patch}}$ to steerable features of type $\frac{C}{8}\rho_{\text{iso}}$ where $C$ is the embedding dimension of the ViT. It is implemented as a convolution over the input image with kernel size and stride equal to the patch size $P$. The convolution kernels are weight sharing constrained to map to features of the different irreps-types in $\frac{C}{8}\rho_{\text{iso}}$, see Appendices E and E.1 for visualizations of the kernels.

Directly after PatchEmbed, we add a learnable positional encoding $\mathbf{e} \in \mathbb{R}^{C \times L}$ to the features. The positional encoding is not constant over tokens, thereby breaking translation equivariance. To be $D_8$ equivariant $\mathbf{e}$ must satisfy

$$\rho_{\text{chan}}(g)\mathbf{x}\rho_{\text{token}}(g)^\mathsf{T} + \mathbf{e} = \rho_{\text{chan}}(g)(\mathbf{x} + \mathbf{e})\rho_{\text{token}}(g)^\mathsf{T} \iff \mathbf{e} = \rho_{\text{chan}}(g)\mathbf{e}\rho_{\text{token}}(g)^\mathsf{T}. \tag{10}$$

In words, the positional encoding at a specific token position $p$ must be a $\rho_{\text{chan}}(g)$-transformed version of the positional encoding at the token position that is permuted to $p$ by $\rho_{\text{token}}(g)$. After adding the positional encoding, we append a learnable class token $[\text{CLS}] \in \mathbb{R}^{C \times 1}$ to the features. To ensure equivariance, we enforce it to be non-zero only in the A1 feature type.

#### 3.3.2 LINEAR LAYERS

Linear layers appear in ViTs both in the MLP block and the MHA block. As mentioned in Section 3.1 and illustrated in Figure 2, equivariant linear layers map between irreps of the same type due to Schur's lemma. This fact was used to construct efficient reflection-equivariant neural networks by Bökman et al. (2025). Here, we use the same approach for octic equivariance.

To re-iterate, for features of type $\rho_{\text{chan}} = \frac{C}{8}\rho_{\text{iso}}$ we consider each token $x \in \mathbb{R}^C$ split into $x_{\text{A1}}, x_{\text{A2}}, x_{\text{B1}}, x_{\text{B2}}, x_{\text{E}}$ of dimensions $C/8, C/8, C/8, C/8$ and $2 \times C/4$. Linear layers map each irrep type to itself, meaning that they are parameterised by four $C/8 \times C/8$ matrices and one $C/4 \times C/4$ matrix, yielding a factor 8 fewer parameters than a general linear layer.

In terms of FLOPs needed to compute the linear layer, $x_i \mapsto W_i x_i$ requires $C/8 \cdot C/8 = C^2/64$ FLOPs for $i \in \{\text{A1, A2, B1, B2}\}$ while due to the "weight-sharing" over the two dimensions in $\rho_E$, it requires $2 \cdot C/4 \cdot C/4 = C^2/8$ FLOPs for $i = \text{E}$. In total we therefore get $16/3 \approx 5.33$ times fewer FLOPs than the $C^2$ required for an ordinary linear layer.

Table 1: **Compute scaling.** We measure the scaling of octic ViTs. The model sizes are taken from Dehghani et al. (2023) and we do not train the largest models as part of this work. The numbers ending in "x" describe the improvement over standard ViT statistics of the corresponding octic ViTs.

| Model | Width | Depth | MLP dim | Heads | throughput | FLOPs | Peak Mem. |
|-------|-------|-------|---------|-------|------------|-------|-----------|
| ViT-L | 1024 | 24 | 4096 | 16 | 1.32x | 4.58x | 2.44x |
| ViT-H | 1280 | 32 | 5120 | 16 | 1.47x | 4.58x | 2.88x |
| ViT-G | 1664 | 48 | 8192 | 16 | 1.91x | 4.88x | 4.36x |
| ViT-e | 1792 | 56 | 15360 | 16 | 2.37x | 5.01x | 5.30x |
| ViT-22B | 6144 | 36 | 24576 | 48 | 3.54x | 5.18x | 5.80x |

### 3.3.3 ACTIVATION FUNCTIONS, LAYER NORM, ATTENTION AND INVARIANTIZATION

A pointwise activation function $\sigma$ can be applied equivariantly after transforming the features from the Fourier domain (multiples of $\rho_{\text{iso}}$), to the spatial domain (multiples of $\rho_{\text{reg}}$), as discussed in Example 3.3. In the spatial domain $\sigma$ can be applied point-wise as this commutes with the permutation representation $\rho_{\text{reg}}$. In our ViTs, following prior work, we apply the GELU activation function.

We implement (token-wise) equivariant layer normalization by transforming each irrep separately to mean 0 followed by division by the total norm over all the irreps.

If two tokens $q$ and $k$ transform according to the same orthogonal representation $\rho_{\text{chan}}$, then $q^{\mathsf{T}}k$ is invariant under $D_8$ since $(\rho_{\text{chan}}(g)q)^{\mathsf{T}}(\rho_{\text{chan}}(g)k) = q^{\mathsf{T}}\rho_{\text{chan}}(g)^{\mathsf{T}}\rho_{\text{chan}}(g)k = q^{\mathsf{T}}k$. This means that the computation of attention logits in ordinary scaled dot-product attention is invariant, so the subsequent weighted sum over value tokens is equivariant.

To output $D_8$ invariant classification predictions, we map from features of type $\frac{C}{8}\rho_{\text{iso}}$ to features of type $C\rho_{\text{A1}}$ and then extract the `[CLS]` token. We ablate different invariantizations to A1-tokens in Appendix F, including linear invariants, triple correlation (Kakarala, 2012; Sanborn & Miolane, 2023), max filtering (Cahill et al., 2024), generators of the ring of invariant polynomials and canonisation of the signal (Kaba et al., 2023). We find that a power spectrum invariantization works well and settle on that for the remainder of the experiments.

### 3.4 COMPUTATIONAL EFFICIENCY

As Transformers scale, the linear layers dominate the execution time (Kaplan et al., 2020). Thus, as ViTs grow, the FLOPs savings will approach those of the linear layer, a reduction of 5.33 times. We plot the FLOPs savings of a ViT block as the embedding dimension increases in Figure 4a. The computational benefits of octic ViTs are more pronounced at scale, and we benchmark the throughput of various ViT sizes from the literature and their octic counterparts in Table 1. We further compare the arithmetic intensity of standard and octic linear layers in Appendix B.

As shown in Table 1, savings in FLOPs do not translate one-to-one to improvements in throughput (images per second) in our current implementation. However, it is still the case that the throughput is greatly improved. Our models are pure PyTorch with `torch.compile`, except for the GELU nonlinearity. We implement a custom Triton (Tillet et al., 2019) kernel that fuses the GELU nonlinearity with the Fourier and inverse Fourier transforms, limiting memory transfers and kernel invocation overhead. While ordinary GELU is pointwise, the new fused Triton kernel is eight to eight points. For extra efficiency, we implement the Fourier transforms by a FFT on $D_8$, described in Appendix A.

## 4 EXPERIMENTS

In this section, we evaluate our octic ViTs on supervised (DeiT III) and self-supervised (DINOv2) training recipes and perform ablations. DeiT III (Touvron et al., 2022) is a popular and highly tuned supervised training recipe for classification and DINOv2 (Oquab et al., 2024) is a state-of-the-art self-supervised method to extract visual features at large scale. All models are trained on ImageNet-1K (Deng et al., 2009; Russakovsky et al., 2015; Recht et al., 2019) following the official implementations. We will release code and pretrained weights on GitHub.

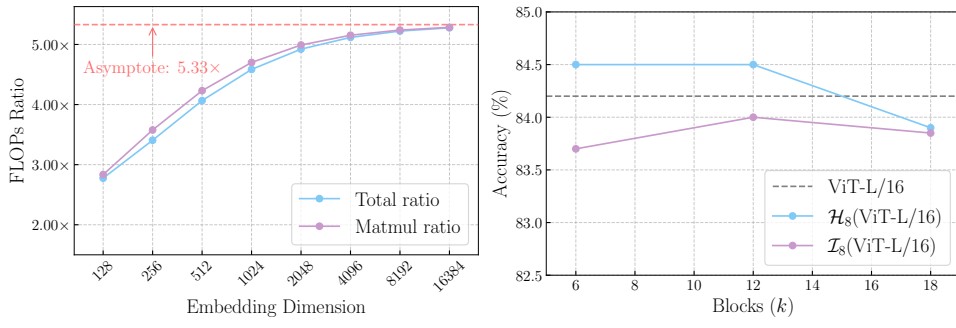

(a) FLOPs ratio vs. embedding dimension  (b) Accuracy vs. number of octic blocks ($k$)

Figure 4: **(a)** Reduction in FLOPs from a non-equivariant Transformer block to an octic-equivariant block vs. embedding dimension. The matmul ratio reflects only matrix multiplications in linear layers and Attention; the total ratio includes all computations. **(b)** The effect of changing the number of octic blocks ($k$) for ViT-L, out of $l = 24$ total blocks.

Table 2: **DeiT III evaluation.** We measure the Top-1 classification accuracy on ImageNet-1K for different model sizes. Our networks are marked with †.

| Model | params ($\times 10^6$) | throughput (im/s) | FLOPs ($\times 10^9$) | Peak Mem. (MB) | Top-1 Acc. ↑ | OOD Rot. $\Delta$Acc.↑ |
|---|---|---|---|---|---|---|
| ViT-H/14 | 632.1 | 569 | 167.8 | 3285 | 84.6 | -12.6 |
| $\mathcal{H}_2$(ViT-H/14) | 474.2 | 640 | 127.4 | 2734 | **85.0** | -13.4 |
| $\mathcal{I}_8$(ViT-H/14)† | 362.3 | 657 | 104.0 | 2249 | 84.7 | **0.0** |
| $\mathcal{H}_8$(ViT-H/14)† | 355.8 | 660 | 102.3 | 2223 | **85.0** | -13.4 |
| $D_2$(ViT-H/14) | 316.1 | 739 | 87.3 | 2100 | 84.4 | -13.1 |
| $D_8$(ViT-H/14)† | 80.5 | 836 | 36.6 | 1141 | 82.0 | **0.0** |
| ViT-L/16 | 304.4 | 1421 | 61.9 | 1557 | 84.2 | -13.4 |
| $\mathcal{H}_2$(ViT-L/16) | 228.3 | 1610 | 46.9 | 1458 | **84.5** | -13.9 |
| $\mathcal{I}_8$(ViT-L/16)† | 175.5 | 1618 | 38.5 | 1210 | 84.0 | **0.0** |
| $\mathcal{H}_8$(ViT-L/16)† | 171.3 | 1615 | 37.7 | 1194 | **84.5** | -13.6 |
| $D_2$(ViT-L/16) | 152.2 | 1868 | 32.2 | 1173 | 83.4 | -14.3 |
| $D_8$(ViT-L/16)† | 39.2 | 1877 | 13.5 | 689 | 79.5 | **0.0** |

## 4.1 DEIT III

We train an array of networks on the supervised task of image classification and compare to the performance reported by Touvron et al. (2022) and Bökman et al. (2025). We find, as illustrated in Table 2, that incorporating octic-equivariant layers provides significant computational savings without sacrificing accuracy. In particular, our $\mathcal{H}_8$(ViT-H/14) model achieves a classification performance of 85.0% compared to the baseline of 84.6% while using only 61% of the FLOPs and matching the performance of the $\mathcal{H}_2$(ViT-H/14) that incorporates flopping ($D_2$) equivariance while being more computational efficient. Similar computational gains are achieved by the invariant model $\mathcal{I}_8$(ViT-H/14), which achieves a classification performance of 84.7%.

In the final column of Table 2, we study the effect of evaluating models on a randomly rotated validation set without training on such augmentations. We find that the invariant model performs equally well while the performance of the remaining models (including $\mathcal{H}_8$) significantly degrade.

## 4.2 DINOv2

As another pretraining task, we consider the DINOv2 recipe and train our own baselines. Results are summarized in Table 3. We find that incorporating octic-equivariant layers maintains or improves downstream classification and segmentation performance while saving FLOPs. In particular, our

Table 3: **DINOv2 evaluation.** We evaluate the frozen DINOv2 features by classification accuracy on ImageNet-1K (IN1K) and segmentation mIoU on ADE20K (Zhou et al., 2017b; 2019) and VOC2012 (Everingham et al., 2010). Our networks are marked with †.

| Model | FLOPs | IN1K (acc.) ↑ | | ADE20K (mIoU) ↑ | | VOC2012 (mIoU) ↑ | |
| --- | --- | --- | --- | --- | --- | --- | --- |
| | $(\times 10^9)$ | linear | $k$-NN | linear | $k$-NN | linear | $k$-NN |
| ViT-H/16 | 127.7 | 81.7 | 81.0 | 34.7 | 30.6 | 70.7 | 60.9 |
| $\mathcal{H}_2$(ViT-H/16) | 97.0 | 81.9 | 80.9 | 34.8 | 30.8 | 70.7 | 61.2 |
| $\mathcal{I}_8$(ViT-H/16)† | 77.7 | 81.9 | 80.9 | 33.9 | 29.2 | 70.6 | 61.2 |
| $\mathcal{H}_8$(ViT-H/16)† | 77.5 | **82.2** | **81.4** | **35.1** | **31.1** | **70.8** | **61.7** |
| ViT-L/16 | 61.9 | 80.9 | 80.5 | 33.2 | 28.4 | 69.1 | 58.1 |
| $\mathcal{H}_2$(ViT-L/16) | 46.9 | **81.3** | 80.7 | 33.4 | 29.0 | 69.3 | 58.0 |
| $\mathcal{I}_8$(ViT-L/16)† | 38.5 | 81.2 | 80.4 | 32.6 | 28.0 | **70.0** | **59.6** |
| $\mathcal{H}_8$(ViT-L/16)† | 37.7 | **81.3** | **80.8** | **33.6** | **29.4** | 69.1 | 59.5 |
| $D_8$(ViT-L/16)† | 13.5 | 72.2 | 71.0 | 23.7 | 17.6 | 53.2 | 42.5 |

invariant network $\mathcal{I}_8$(ViT-H/16) matches the downstream performance of the baseline while using only 61% of the FLOPs and $\mathcal{H}_8$(ViT-H/16) slightly improves performance with similar savings.

In Appendix G.3, we further investigate the performance of our invariant model and evaluate it on white blood cell classification, a task that lacks a canonical orientation. The invariant model $\mathcal{I}_8$(ViT-L/16) outperforms the baseline on most evaluated metrics.

### 4.3 ABLATIONS

**Impact of equivariance.** We study the effect of equivariance by replacing the kernel constrained equivariant patch embed by an arbitrarily linear mapping while keeping the rest of the architecture the same as for $\mathcal{H}_8$. In principle, the model can learn to be equivariant. We evaluate ViT-B using the DeiT III recipe and achieve an accuracy of $82.4$ compared to $83.0$ for $\mathcal{H}_8$(ViT-B). The results suggest that equivariance yields higher accuracy than arbitrary mappings with the same block-diagonal structure (in addition to providing steerable features that can be useful in downstream applications).

**Number of octic blocks.** We experiment with incorporating non-equivariant blocks following Weiler & Cesa (2019). We include networks where the first $k$ of the ViT blocks are octic and the remaining $l - k$ are standard blocks. Figure 4b ablates different values of $k$ for ViT-L ($l = 24$). We find that $k = \frac{l}{2}$ strikes a good balance between computational efficiency and representational power and use this throughout the paper. Note that $\mathcal{I}_8$ performs worse for small $k$ due to early invariantization.

We find that the $\mathcal{H}_8$-networks typically outperform $\mathcal{I}_8$-networks. This usefulness of symmetry breaking corroborates findings in prior work, e.g. (Weiler & Cesa, 2019; Vadgama et al., 2025). The intuition is that ImageNet is not a rotationally invariant dataset, so it leads to improved performance to let the network use the fact that the images are upright as extra information for solving the classification task.

## 5 LIMITATIONS

In this work, we limit our scope to an extensive study of the $D_8$ group and leave larger dihedral groups for future work, we provide a theoretical discussion regarding scaling to larger dihedral groups in Appendix D for reference. We follow baseline training recipes without hyperparameter tuning, and we do not conduct an extensive ablation study of the share of features per irrep or scale the size beyond ViT-H. We aim to explore this in future work. Moreover, we do not realize the full throughput potential of our octic layers, as illustrated by lower throughput gains than FLOPs savings. This point is discussed in terms of arithmetic intensity in Appendix B. Our results show promise in continuing to work in the direction of equivariance for computational efficiency.

## 6  CONCLUSION

We introduced octic-equivariant ViT layers that, when incorporated, maintain accuracy while significantly reducing computational complexity. We validated our proposed architectures by their effectiveness in both supervised and self-supervised learning, and conducted ablation studies to isolate the effect of invariantization, equivariance, and the number of octic blocks. In particular, we achieved an approximate 40% reduction in FLOPs for ViT-H without sacrificing accuracy, positioning octic ViTs as a strong addition to the catalog of vision architectures.

## REPRODUCIBILITY STATEMENT

To ensure reproducibility, we have chosen to evaluate our work on publicly available datasets (Deng et al., 2009; Everingham et al., 2010; Zhou et al., 2017b) and industry-standard training recipes (Touvron et al., 2021; Oquab et al., 2024). We provide additional details on the experiments in Appendix G. The architecture and main layers to construct our proposed octic ViTs are detailed in Section 3. We will also release code and pretrained weights on GitHub.

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

# A FOURIER TRANSFORM FOR $D_8$

## A.1 IMPLEMENTATION

The Inverse Fourier Transform, i.e., changing basis from the Isotypical representation $\rho_{\text{iso}}$ to the Regular representation $\rho_{\text{reg}}$, can be written in the case of $D_8$ as

$$
Q_{\text{reg}} = \frac{\sqrt{2}}{4} \begin{pmatrix}
1 & 1 & 1 & 1 & 1 & 1 & 1 & -1 \\
1 & 1 & -1 & -1 & 1 & -1 & -1 & -1 \\
1 & 1 & 1 & 1 & -1 & -1 & -1 & 1 \\
1 & 1 & -1 & -1 & -1 & 1 & 1 & 1 \\
1 & -1 & 1 & -1 & -1 & 1 & -1 & -1 \\
1 & -1 & -1 & 1 & -1 & -1 & 1 & -1 \\
1 & -1 & 1 & -1 & 1 & -1 & 1 & 1 \\
1 & -1 & -1 & 1 & 1 & 1 & -1 & 1
\end{pmatrix}.
\tag{11}
$$

In practice, we use a fast Triton-compiled implementation of the mapping $x \mapsto Q_{\text{reg}}x$ as shown in Listing 1, and similarly for $x \mapsto Q_{\text{reg}}^{\mathsf{T}}x$.

Listing 1: Python implementation of $Q_{\text{reg}}$.

```python
import math
SQRT2_OVER_4 = math.sqrt(2) / 4

def isotypical_to_regular(
    x_A1, x_A2, x_B1, x_B2, x_E11, x_E12, x_E21, x_E22
):
    a = x_A1 + x_A2
    b = x_A1 - x_A2
    c = x_B1 + x_B2
    d = x_B1 - x_B2
    e = x_E11 + x_E12
    f = x_E11 - x_E12
    g = x_E21 + x_E22
    h = x_E21 - x_E22
    apc = a + c
    amc = a - c
    bpd = b + d
    bmd = b - d
    eph = e + h
    emh = e - h
    fpg = f + g
    fmg = f - g
    return (
        SQRT2_OVER_4 * (apc + eph),
        SQRT2_OVER_4 * (amc + fmg),
        SQRT2_OVER_4 * (apc - eph),
        SQRT2_OVER_4 * (amc - fmg),
        SQRT2_OVER_4 * (bpd - fpg),
        SQRT2_OVER_4 * (bmd - emh),
        SQRT2_OVER_4 * (bpd + fpg),
        SQRT2_OVER_4 * (bmd + emh)
    )
```

## A.2 TIME COMPLEXITY

The time complexity of FFT/iFFT is linear with respect to $C$ and log-linear with respect to the order of the dihedral group, but since the group remains constant in this work, we focus the study on the time complexity with respect to the embedding dimension $C$. The complexity of iFFT→GELU→FFT is thus linear in $C$, which is the same as the complexity of just using GELU. In contrast, the complexity of linear layers is quadratic in $C$, so savings in the linear layers will more than compensate additional computation in the nonlinearity when $C$ grows.

Table 4: **Time complexity of non-linearity.** Comparing the time (in $\mu$s) of GELU and the MLP block for standard and octic implementations. The extra computations needed for the non-linearity are noticed at small scale but as $C$ grows the savings from the linear layers dominate.

| Model | $C = 256$ | $C = 512$ | $C = 1024$ | $C = 2048$ |
|---|---|---|---|---|
| GELU | 44.8 | 82.8 | 155.8 | 302.6 |
| $\text{GELU}_{D_8}$ | 46.1 | 84.7 | 158.8 | 307.0 |
| MLP | 128.4 | 324.2 | 1053.3 | 3644.9 |
| $\text{MLP}_{D_8}$ | 142.4 | 244.4 | 471.4 | 1123.4 |

We benchmark actual runtime on an A100 GPU. We compare the two non-linearities and full MLP blocks (Linear$(C, 4C)$ - Nonlinearity - Linear$(4C, C)$), for various $C$. Times are in $\mu$s, averaged over 1000 runs of a forward pass with batch size 32. Non-linearities are run on embedding dimension $4C$. We report the results in Table 4.

Consistent with previous results, we find that the equivariant linear layers pay off more with increasing embedding dimension. The extra computations needed for the non-linearity give only a few percent overhead while the savings from the linear layers provide substantial performance gains.

# B  ARITHMETIC INTENSITY

The arithmetic intensity measures FLOPs per transferred byte and can be compared with the FLOPs per bandwidth of a given device to obtain a bound on the maximum achievable throughput (Austin et al., 2025). The arithmetic intensities are $\frac{2BCF}{P(BC+CF+BF)}$ and $\frac{2BCF/5.33}{P(BC+CF/8+BF)}$ for the standard and octic linear layers, respectively, where $B$ is the batch size in tokens, $C$ is the input dimension, $F$ is the output dimension and $P$ is the precision in bytes. This means that the octic and ordinary layers scale differently. At large scale, not only FLOPs are improved by octic layers but also arithmetic intensity. For instance, for $B = 196$ (one image worth of tokens), $P = 2$ and $F = 4C$ (a typical MLP expansion factor) one can calculate that ordinary linear layers have higher arithmetic intensity up to $C \approx 3200$, whereas octic linear layers have higher arithmetic intensity at larger dimensions. For the experiments in this paper, we are not able to scale to such large dimensions, but still get throughput benefits due to savings in FLOPs, as shown in Table 1.

For larger batch sizes (say, 64 images worth of tokens), standard linear layers typically have larger arithmetic intensity than octic linear layers, because the terms without $B$ in the denominator become negligible. However at sufficiently large $C$ both standard and octic linear layers are compute bound and hence what matters for throughput is the total number of FLOPs, making octic linear layers 5.33 times faster. We make this discussion more precise in the next section.

## B.1  THROUGHPUT BENCHMARK FOR LINEAR LAYERS

As mentioned in Section A.2, the main bottleneck to achieve throughput gains closer to the actual FLOP reductions in octic ViTs is that the linear layers are not performing sufficiently well for low feature dimensions $C$.

To more clearly illustrate the bottleneck and tie it to the arithmetic intensity (AI) of the layers we have conducted a simple throughput benchmark for only linear layers. We include standard linear layers, octic linear layers and block-diagonal layers with eight equal size blocks, all from $C$ to $F = 4C$ channels. The block-diagonal layers have arithmetic intensity $\frac{2BCF/8}{P(BC+CF/8+BF)}$.

We benchmark with batch size $64 \cdot 196$ (i.e. 64 images worth of tokens) on an A100-80GB GPU. The advertised maximum achievable performance using bf16 or fp16 precision is 312 TFLOPs/s. We use bf16 precision when training the networks in this paper. For reference we also include results in fp32 precision, where the advertised maximum achievable performance is 19.5 TFLOPs/s. The ridge point for when computations become theoretically compute bound is at 156 FLOPs/byte for the lower precision and 10 FLOPs/byte for the higher precision. We see in Table 5 that for non-compute

Table 5: **Benchmarking linear layers.** Performance of three different types of linear layers over different precisions and different amount of channels.

| Linear layer | Precision | Channels | AI (FLOPs/byte) | Time (ms) | Perf. (TFLOPs/s) | Speedup |
|---|---|---|---|---|---|---|
| Standard | fp16 | $1024 \rightarrow 4096$ | 770 | 0.47 | 230 | 1.0 |
| Octic | fp16 | $1024 \rightarrow 4096$ | 150 | 0.19 | 110 | 2.6 |
| Eight blocks | fp16 | $1024 \rightarrow 4096$ | 100 | 0.13 | 100 | 3.6 |
| Standard | fp16 | $8192 \rightarrow 32768$ | 4300 | 24 | 280 | 1.0 |
| Octic | fp16 | $8192 \rightarrow 32768$ | 1200 | 5.3 | 240 | 4.6 |
| Eight blocks | fp16 | $8192 \rightarrow 32768$ | 770 | 3.4 | 250 | 7.2 |
| Standard | fp32 | $1024 \rightarrow 4096$ | 380 | 5.6 | 19 | 1.0 |
| Octic | fp32 | $1024 \rightarrow 4096$ | 80 | 1.2 | 17 | 4.8 |
| Eight blocks | fp32 | $1024 \rightarrow 4096$ | 50 | 0.76 | 17 | 7.4 |
| Standard | fp32 | $8192 \rightarrow 32768$ | 2200 | 360 | 19 | 1.0 |
| Octic | fp32 | $8192 \rightarrow 32768$ | 580 | 67 | 19 | 5.33 |
| Eight blocks | fp32 | $8192 \rightarrow 32768$ | 380 | 45 | 19 | 7.95 |

Table 6: **Training speed**. We measure the training speed (forward + backward) when using octic ViTs in different training pipelines. We report the throughput (images/second) and the speed gain compared to the baseline. We run on an A100-80GB in mixed precision.

| Model | DeiT III | | DINOv2 | |
|---|---|---|---|---|
| | Throughput | Gain | Throughput | Gain |
| ViT-H | 180 | $1.00\times$ | 94 | $1.00\times$ |
| $\mathcal{I}_8(\text{ViT-H})$ | 207 | $1.15\times$ | 114 | $1.21\times$ |
| $\mathcal{H}_8(\text{ViT-H})$ | 209 | $1.16\times$ | 115 | $1.22\times$ |
| $D_8(\text{ViT-H})$ | 257 | $1.43\times$ | 142 | $1.51\times$ |
| ViT-L | 454 | $1.00\times$ | 184 | $1.00\times$ |
| $\mathcal{I}_8(\text{ViT-L})$ | 510 | $1.12\times$ | 210 | $1.14\times$ |
| $\mathcal{H}_8(\text{ViT-L})$ | 512 | $1.13\times$ | 214 | $1.16\times$ |
| $D_8(\text{ViT-L})$ | 564 | $1.24\times$ | 232 | $1.26\times$ |

bound layers such as octic and block-diagonal linears at $C = 1024$ in fp16, the performance is far below the compute bound layers.

## C  TRAINING SPEED COMPARISONS

In the main manuscript only inference speed is measured (e.g. see Tables 2 and 1). In Table 6, we show that inference throughput gains also translate into training (forward and backward) improvements by measuring the training throughput (images/second).

## D  LARGER DIHEDRAL GROUPS

In this section, we briefly sketch the generalizaton from the octic group to larger dihedral groups. We will consider the dihedral group $D_{4n}$ of order $4n$, meaning $180/n$ degree rotations and reflections. This group has 4 one dimensional irreps and $n - 1$ two-dimensional irreps. The theoretical discussion from Section 3 carries over almost as is. The real irreps are all realizable over the complex numbers, so Schur's lemma implies that linear layers from $\frac{C}{4n}$ copies of the regular representation to itself have 4 blocks of size $\frac{C}{4n}$ and $n - 1$ twice repeated blocks of size $\frac{C}{2n}$. In Figure 2 we see the case $n = 2$ for the octic group.

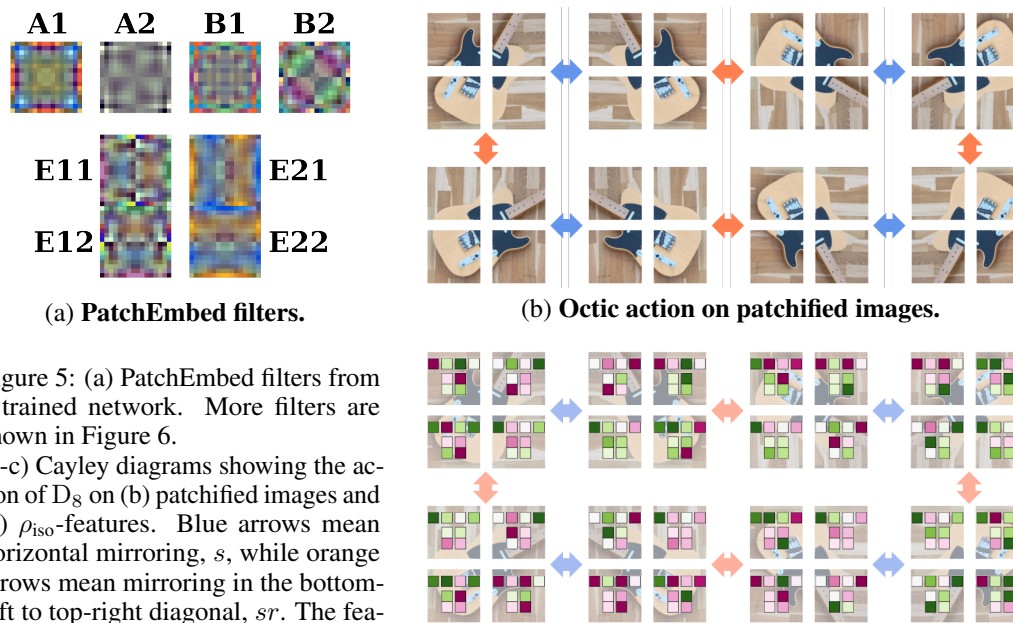

(a) **PatchEmbed filters.**

(b) **Octic action on patchified images.**

(c) **Octic action on $\rho_{\text{iso}}$-features.**

Figure 5: (a) PatchEmbed filters from a trained network. More filters are shown in Figure 6.
(b-c) Cayley diagrams showing the action of $D_8$ on (b) patchified images and (c) $\rho_{\text{iso}}$-features. Blue arrows mean horizontal mirroring, $s$, while orange arrows mean mirroring in the bottom-left to top-right diagonal, $sr$. The features were obtained by applying the filters in (a) to the patches in (b).

The number of multiplications required for an equivariant linear layer is then $4 \cdot (\frac{C}{4n})^2 + 2 \cdot (n-1) \cdot (\frac{C}{2n})^2 = \frac{C^2(2n-1)}{4n^2}$. The compute savings relative to an ordinary linear layer are therefore $\frac{4n^2}{2n-1}$ times fewer FLOPs.

The arithmetic intensity for an equivariant linear layer from $C$ to $F$ channels is $\frac{2BCF(2n-1)/(4n^2)}{P(BC+CF/(4n)+BF)}$. Just as discussed for the octic group in Section B, whether the full FLOP saving of the linear layers can be realized or not depends on $B, C, F, P$ and the specific hardware used.

# E    VISUALIZATIONS

We visualize the action of the octic group on images and on $\rho_{\text{iso}}$-features in Figure 5.

## E.1    LEARNED FILTERS

The PatchEmbed layer contains filters mapping the input from 3 channel dimensions to $C$ embedding dimensions. To illustrate the learned filters, we take inspiration from Dosovitskiy et al. (2021) and visualize the first 16 principal components. In contrast to regular ViTs, we have six different learned filters corresponding to the five irreps. Four for the one-dimensional irrep and two for the E irrep (due to its multiplicity). The results are illustrated in Figure 6. Interestingly, the learned filters look qualitatively different between the two learning methods. This is similar to how the learned filters of baseline DINOv2 and DeiT III look qualitatively different. DINOv2 training appears to produce more high frequency patterns while DeiT III gives clearer patterns. For the invariant irrep (A1), it appears that the DeiT III training produces a spherical pattern.

# F    INVARIANTIZATION

There are multiple options to produce $D_8$ invariant features, i.e. mapping tokens of type $\rho_{\text{chan}} = \frac{C}{8}\rho_{\text{iso}}$ to features of type $C\rho_{\text{A1}}$ (here denoted in short as *invariantization*). We let $\psi$ be a function mapping from features of type $\frac{C}{8}\rho_{\text{iso}}$ to features of type $\frac{KC}{8}\rho_{\text{A1}}$ for some $K$ which can be larger or smaller than 8. These $KC/8$ dimensions are then mapped through a small MLP to $C$ dimensions again.

**Linear Invariant (Linear).** The linear invariant simply extracts the invariant irrep. Here, $K = 1$.

$$\psi(x_{A1}, x_{A2}, x_{B1}, x_{B2}, x_{E11}, x_{E12}, x_{E21}, x_{E22}) = x_{A1}. \tag{12}$$

**Triple Correlation (Triple Corr.).** The triple correlation method (Sanborn & Miolane, 2023; Kakarala, 2012) extracts a complete set of third order homogeneous polynomial invariants from a signal over $D_8$. We computed a basis for all third order invariant homogeneous polynomials using Macaulay2 (Grayson & Stillman; Ferraro et al., 2024) and use the basis elements as invariant. Here, $K = 15$.

$$\psi(x_{A1}, x_{A2}, x_{B1}, x_{B2}, x_{E11}, x_{E12}, x_{E21}, x_{E22})$$

$$= \Big( x_{A1}^3, x_{A1}(x_{E21}^2 + x_{E22}^2), x_{A1}(x_{E11}x_{E21} + x_{E12}x_{E22}), x_{A1}(x_{E11}^2 + x_{E12}^2), x_{A1}x_{B2}^2, x_{A1}x_{B1}^2, x_{A1}x_{A2}^2,$$

$$x_{B2}x_{E21}x_{E22}, x_{B2}x_{E12}x_{E21} + x_{B2}x_{E11}x_{E22}, x_{B2}x_{E11}x_{E12}, x_{B1}x_{E21}^2 - x_{B1}x_{E22}^2,$$

$$x_{B1}x_{E11}x_{E21} - x_{B1}x_{E12}x_{E22}, x_{B1}x_{E11}^2 - x_{B1}x_{E12}^2, x_{A2}x_{E12}x_{E21} - x_{A2}x_{E11}x_{E22}, x_{A2}x_{B1}x_{B2} \Big) \tag{13}$$

**Power spectrum.** A common invariant is the power spectrum. We use the following variant, with $K = 6$.

$$\psi(x_{A1}, x_{A2}, x_{B1}, x_{B2}, x_{E1}, x_{E2}) = (x_{A1}, |x_{A2}|, |x_{B1}|, |x_{B2}|, \|x_{E1}\|, \|x_{E2}\|) \tag{14}$$

**Polynomial.** Similar to the triple correlation, we can consider a polynomial basis for the full invariant ring. This was computed using Macaulay2, yielding $K = 32$.

$$\psi(x_{A1}, x_{A2}, x_{B1}, x_{B2}, x_{E11}, x_{E12}, x_{E21}, x_{E22})$$

$$= \Big( x_{A1}, x_{E21}^2 + x_{E22}^2, x_{E11}x_{E21} + x_{E12}x_{E22}, x_{E11}^2 + x_{E12}^2, x_{B2}^2, x_{B1}^2, x_{A2}^2,$$

$$x_{B2}x_{E21}x_{E22}, x_{B2}x_{E12}x_{E21} + x_{B2}x_{E11}x_{E22}, x_{B2}x_{E11}x_{E12}, x_{B1}x_{E21}^2 - x_{B1}x_{E22}^2,$$

$$x_{B1}x_{E11}x_{E21} - x_{B1}x_{E12}x_{E22}, x_{B1}x_{E11}^2 - x_{B1}x_{E12}^2, x_{A2}x_{E12}x_{E21} - x_{A2}x_{E11}x_{E22},$$

$$x_{A2}x_{B1}x_{B2}, x_{E21}^4 + x_{E22}^4, x_{E11}x_{E21}^3 + x_{E12}x_{E22}^3, x_{E11}^2x_{E21}^2 + x_{E12}^2x_{E22}^2,$$

$$x_{E11}^3x_{E21} + x_{E12}^3x_{E22}, x_{E11}^4 + x_{E12}^4,$$

$$x_{B1}x_{B2}x_{E12}x_{E21} - x_{B1}x_{B2}x_{E11}x_{E22}, x_{A2}x_{B2}x_{E21}^2 - x_{A2}x_{B2}x_{E22}^2,$$

$$x_{A2}x_{B2}x_{E11}x_{E21} - x_{A2}x_{B2}x_{E12}x_{E22}, x_{A2}x_{B2}x_{E11}^2 - x_{A2}x_{B2}x_{E12}^2,$$

$$x_{A2}x_{B1}x_{E21}x_{E22}, x_{A2}x_{B1}x_{E12}x_{E21} + x_{A2}x_{B1}x_{E11}x_{E22}, x_{A2}x_{B1}x_{E11}x_{E12},$$

$$x_{A2}x_{E21}^3x_{E22} - x_{A2}x_{E21}x_{E22}^3, x_{A2}x_{E12}x_{E21}^3 - x_{A2}x_{E11}x_{E22}^3,$$

$$x_{A2}x_{E11}x_{E12}x_{E21}^2 - x_{A2}x_{E11}x_{E12}x_{E22}^2, x_{A2}x_{E11}^2x_{E12}x_{E21} - x_{A2}x_{E11}x_{E12}^2x_{E22},$$

$$x_{A2}x_{E11}^3x_{E12} - x_{A2}x_{E11}x_{E12}^3 \Big) \tag{15}$$

**Max filtering.** We follow Cahill et al. (2024) and implement a version of their max filtering invariant. For this, we have a set of $2C$ learnable $C$-dimensional tokens $\mathbf{y} \in \mathbb{R}^{2C \times C}$, and the $2C$ invariants are given by

$$\psi(x) = \oplus_{k=1}^{2C} \max_{g \in D_8} \langle \mathbf{y}_k, \rho_{\text{chan}}(g)x \rangle. \tag{16}$$

**Canonisation.** Similar to the max filtering approach, we implement a canonisation where we have a single learnable $C$-dimensional reference token $y$ and compute the $C$-dimensional invariant as

$$\psi(x) = \rho_{\text{chan}}\left(\text{argmax}_{g \in D_8} \langle y, \rho_{\text{chan}}(g)x \rangle\right) x. \tag{17}$$

We conduct a study of the effect of these different invariantization methods. A priori, max filtering and canonisation should be more expressive than the others as they are the only invariants considered

Table 7: **Invariantization Ablation.** Comparing classification accuracy using different invariantization methods on ImageNet-1K using the DeiT III training recipe for 400 epochs for $D_8$ (ViT-L/16).

| | Top-1 | Top-5 |
|---|---|---|
| Linear | 79.2 | 94.7 |
| Polynomial | 79.4 | 94.8 |
| Triple Corr. | 76.8 | 93.7 |
| Canonisation | **79.5** | **94.9** |
| Max Filtering | 77.9 | 93.9 |
| Power Spec. | **79.5** | **94.9** |

Table 8: **Hyperparameters used in experiments.** Collection of the most important hyperparameters used. For the full set we refer to the original implementations.

(a) DeiT III hyperparameters

| Model | ViT-L/16 | ViT-H/14 |
|---|---|---|
| Batch size | $128 \times 16$ | $64 \times 32$ |
| Optimizer | LAMB | LAMB |
| LR | $3 \times 10^{-3}$ | $3 \times 10^{-3}$ |
| LR decay | cosine | cosine |
| Weight decay | 0.02 | 0.02 |
| Training epochs | 400 | 400 |
| Warmup epochs | 5 | 5 |
| Stoch. Depth | 0.4 | 0.5 |
| Repeated Aug | ✓ | ✓ |
| Gradient Clip. | 1.0 | 1.0 |
| Mixup alpha | 0.8 | 0.8 |
| Cutmix alpha | 1.0 | 1.0 |
| ColorJitter | 0.3 | 0.3 |
| Loss | BCE | BCE |

(b) DINOv2 hyperparameters

| Model | ViT-L/16 | ViT-H/16 |
|---|---|---|
| Batch size | $64 \times 16$ | $32 \times 32$ |
| Optimizer | AdamW | AdamW |
| Base LR | $4 \times 10^{-3}$ | $4 \times 10^{-3}$ |
| Init layer scale | $1.10^{-5}$ | $1.10^{-5}$ |
| Weight decay | 0.04 | 0.04 |
| Training steps | 125K | 125K |
| Warmup steps | 12.5K | 12.5K |
| Drop path | 0.3 | 0.3 |
| K | 65536 | 65536 |
| Gradient Clip. | 3.0 | 3.0 |
| Init EMA momentum | 0.8 | 0.8 |
| Koleo loss weight | 0.1 | 0.1 |
| iBOT sample prob. | 0.5 | 0.5 |
| iBOT mask ratio | 0.1-0.5 | 0.1-0.5 |

here that are able to preserve the relative phase information coming from different phase in different copies of $\rho_{iso}$. We train $D_8$(ViT-L/16) on ImageNet-1K following the DeiT III recipe. The results are presented in Table 7. The conclusion is that the simple power spectrum invariant works well, and so we select it as our invariantization of choice in the remainder of the experiments.

# G    EXPERIMENTAL SETTING

## G.1    DEIT III

We train for 400 epochs on ImageNet-1K with an effective batch size of 2048 following Touvron et al. (2022) and Bökman et al. (2025). We compare to the figures reported in the respective papers and thus only train the octic ViTs. The training recipe includes heavy data augmentation (e.g. cutmix, mixup and color jitter) and uses the deprecated NVIDIA `Apex` library. Training is done in mixed precision with the `lamb` optimizer. The most important hyperparameters are summarized in Table 8a. For more details, we refer to (Touvron et al., 2022; Bökman et al., 2025) and the official repo used for reproduction `https://github.com/facebookresearch/deit`.

Out of distribution (OOD) rotation evaluation simply adds a random 90 degree rotation to the validation set of IN1K and computes the classification accuracy on the randomly rotated dataset. Note, as the publicly available weights are trained for 800 epochs (whereas we compare to the figures reported for 400 epochs in the original paper), we compute the OOD $\Delta$ on these weights.

Table 9: **DINOv2 additional evaluation.** We further evaluate the frozen DINOv2 features by classification accuracy on iNaturalist2021 (Van Horn et al., 2021) and Places365 (Zhou et al., 2017a).

| Model | FLOPs $(\times 10^9)$ | iNaturalist2021 ↑ | | Places365 ↑ | |
|---|---|---|---|---|---|
| | | linear | $k$-NN | linear | $k$-NN |
| ViT-H/16 | 127.7 | 81.7 | 81.0 | 34.7 | 30.6 |
| $\mathcal{I}_8$(ViT-H/16) | 77.7 | 81.9 | 80.9 | 33.9 | 29.2 |
| $\mathcal{H}_8$(ViT-H/16) | 77.5 | **82.2** | **81.4** | **35.1** | **31.1** |
| ViT-L/16 | 61.9 | 80.9 | 80.5 | 33.2 | 28.4 |
| $\mathcal{I}_8$(ViT-L/16) | 38.5 | 81.2 | 80.4 | 32.6 | 28.0 |
| $\mathcal{H}_8$(ViT-L/16) | 37.7 | **81.3** | **80.8** | **33.6** | **29.4** |

## G.2 DINOv2

We closely follow the implementation in the original paper, only modifying to train in BF16 instead of FP16 for greater stability, and follow the same evaluation protocol for classification. Lacking an official reproduction of the segmentation protocol in DINOv2, we opt for the evaluation protocol created by Darcet et al. (2025) for semantic segmentation on ADE20K and VOC2012. In contrast to the original DINOv2 paper, we decide to limit our study to ViT-L and ViT-H, the latter of which was not included in the original paper. We opt for the larger patch size of $P = 16$ for all our DINOv2 models to save computational resources. Note, this is the reason why we report fewer FLOPs for our DINOv2 ViT-H models than their DeiT III counterpart (which use $P = 14$).

We train on ImageNet-1K for 125K steps with an effective batch size of 1024 using the `adamw` optimizer. We train our own baselines for fair comparison (to obtain checkpoints after only training on IN1K). The training progression of the ViT-L/16 family can be visualized in Figure 7. The most important hyperparameters are summarized in Table 8b. For exact details about the configuration of hyper parameters we refer to the base configs in the DINOv2 repo `https://github.com/facebookresearch/dinov2`.

We do not implement specific hardware efficient layers for DINOv2 training and instead opt for the standard octic layers that are `timm` compatible. As such, the octic layers do not leverage `NestedTensorBlock` and training speedups associated with `xFormers`. This choice does not impact speed on downstream tasks but slightly decreases pre-training speed.

We extend the evaluation of DINOv2 (trained on ImageNet-1K) to two popular classification datasets and report the performance in Table 9. We find, once again, that our networks achieve similar or better performance than the baseline while using substantially fewer FLOPs.

## G.3 DINOBLOOM

We investigate the performance of our invariant model on white blood cell classification. We follow the procedure of DinoBloom (Koch et al., 2024) and report our results for ViT-L in Table 11. We find that the invariant model $\mathcal{I}_8$(ViT-L/16) outperforms the baseline on most evaluated metrics. We tried evaluating on a rotated test set and found negligible change in performance for the baseline (the invariant model inherently has no change in performance, similar to the last column of Table 2).

For the details of the experiment, we closely follow the training and evaluation protocol of Koch et al. (2024). In particular, we finetune our DINOv2 checkpoints for 4K iterations (taking approx. 1 hour on an 8× A100-40GB node) and evaluate on a hold-out split of the Bone Marrow Cytomorphology (BMC) (Matek et al., 2021) dataset. However, we limit our finetuning datasets to the datasets presented in Table 10. Note, we follow the same datasplit of BMC as in DinoBloom and thus also refrain from training on that part.

Table 10: **Hematology datasets**. Dataset mixture for DinoBloom finetuning.

| Datasets | Modality | Images |
|----------|----------|--------|
| BMC | Bone Marrow | 171K |
| AML Hehr | Blood | 102K |
| APL | Blood | 26K |
| AML Matek | Blood | 18K |
| Acevedo | Blood | 17K |
| Raabing WBC | Blood | 10K |
| **Total** | | **354k** |

Table 11: **Hematology finetuning.** White blood cell classification performance on BMC dataset with 21 highly imbalanced classes after finetuning on hematology data following DinoBloom.

| Model | 1-$k$-NN | | | 20-$k$-NN | | | Lin. probe | | |
|-------|------|-----|------|------|-----|------|------|-----|------|
|       | wF1  | Acc | bAcc | wF1  | Acc | bAcc | wF1  | Acc | bAcc |
| ViT-L/16 | 78.0 | **78.0** | 57.6 | 83.1 | 83.6 | 54.9 | 84.6 | 84.7 | **62.2** |
| $\mathcal{I}_8$(ViT-L/16) | **78.1** | **78.0** | **61.3** | **83.5** | **83.9** | **55.7** | **85.0** | **85.2** | 61.9 |

### G.4 ROTATION ERROR

In Table 12, we show the difference in classification performance under $45°$ rotations. The $\mathcal{I}_8$ models are substantially more robust to rotations.

### G.5 GENERAL SETTINGS

**Software versioning.** We utilize the PyTorch (Paszke et al., 2019) and the timm (Wightman, 2019) libraries for our experiments. We run the same versioning as our benchmarks. For all other experiments, we use Python 3.11.9 and PyTorch 2.6.0 with CUDA 11.8.

**Model sizes.** The model sizes referred to in the paper adhere to the standard terminology used by Wightman (2019); Dosovitskiy et al. (2021). If we denote the shape by a tuple of (depth, width, attention heads), ViT-L has shape (24, 1024, 16) and ViT-H has shape (32, 1280, 16). Both use MLP dimension four times the size of the embedding dimension (commonly referred to as MLP ratio).

**Calculating throughput.** Throughput and peak memory are measured on a single A100-80GB GPU with batch size fixed to 64 using `torch.compile`, FlashAttention (Dao, 2024), and mixed precision. The throughput only measures forward passes with no gradients. Moreover, we utilize 10 warm-up iterations and then average over 100 runs (Bökman et al., 2025). Peak memory is measured with PyTorch's device memory allocation monitor.

**Counting FLOPs.** We count the number of FLOPs using `fvcore.nn.FlopCountAnalysis` `https://github.com/facebookresearch/fvcore`. FLOPs are normalized with respect to the batch size (i.e. we measure FLOPs/image). We acknowledge that the term FLOPs often leads to confusion. We adopt the terminology of prior work (Touvron et al., 2022; Bökman et al., 2025) and the `fvcore` library for FLOPs, though, strictly speaking, this refers to MACs (as a factor of two is omitted).

### G.6 COMPUTE RESOURCES

Table 13 provides information on an exact account of the computing resources used. Failed or discarded results are not included. In total, we used around 20k A100-40GB hours for our main results. Note, this should not be considered a fair throughput comparison as not all speed optimizations were implemented at the time of the initial experiments. Notably, the DINOv2 comparison is not fair due to the use of a nested kernel for pre-training for the baseline but not for the equivariant networks.

Table 12: **Classification performance under 45° rotations.** In addition to the 90° rotation error we evaluate in Table 2, we show the difference in classification performance from upright images to 45° rotations.

| Model | $\Delta$ acc. |
|---|---|
| ViT-H/16 | -13.1 |
| $\mathcal{I}_8$(ViT-H/16) | -5.4 |
| $\mathcal{H}_8$(ViT-H/16) | -13.5 |
| ViT-L/16 | -13.3 |
| $\mathcal{I}_8$(ViT-L/16) | -5.7 |
| $\mathcal{H}_8$(ViT-L/16) | -13.6 |

Table 13: **Compute accounting.** Exact account of hardware usage for the main experiments. It does not constitute a fair speed comparison between the different models. For a fair training time comparison, see Table 6. Resource unit (RU) is measured in A100 equivalent hours where an A40 hour costs 0.54 units.

| Experiment | Model | GPUs | Time | RU |
|---|---|---|---|---|
| DeiT III | $\mathcal{I}_8$(ViT-H/14) | 32×A100-40GB | 58h | 1856 |
| DeiT III | $D_8$(ViT-H/14) | 32×A100-40GB | 57h | 1824 |
| DeiT III | $\mathcal{H}_8$(ViT-H/14) | 32×A100-40GB | 55h | 1760 |
| DeiT III | $\mathcal{H}_8$(ViT-L/16) | 16×A100-40GB | 39h | 624 |
| DeiT III | $\mathcal{I}_8$(ViT-L/16) | 16×A100-40GB | 37h | 592 |
| DeiT III | $D_8$(ViT-L/16) | 16×A100-40GB | 36h | 576 |
| Ablation: Invarisation | $\mathcal{H}_8$(ViT-L/16) | 16×A100-40GB | 240h | 3840 |
| Ablation: Hybridisation | $\mathcal{I}_8$(ViT-L/16) | 16×A100-40GB | 122h | 1952 |
| Ablation: Hybridisation | $\mathcal{H}_8$(ViT-L/16) | 16×A100-40GB | 120h | 1920 |
| DINOv2 | $\mathcal{H}_8$(ViT-H/16) | 32×A100-40GB | 33h | 1056 |
| DINOv2 | $\mathcal{I}_8$(ViT-H/16) | 32×A100-40GB | 30h | 960 |
| DINOv2 | $D_8$ (ViT-L/16) | 32×A100-40GB | 23h | 736 |
| DINOv2 | ViT-H/16 | 32×A100-40GB | 21h | 672 |
| DINOv2 | $\mathcal{I}_8$(ViT-L/16) | 32×A100-40GB | 19h | 608 |
| DINOv2 | $\mathcal{H}_8$(ViT-L/16) | 32×A100-40GB | 19h | 608 |
| DINOv2 | ViT-L/16 | 16×A100-40GB | 24h | 384 |
| DinoBloom | $\mathcal{I}_8$(ViT-L/16) | 8×A100-40GB | 46min | 6 |
| DinoBloom | ViT-L/16 | 8×A100-40GB | 37min | 5 |
| **Total** | | | | **19979** |

## H  LICENSES

**Data.** All our models are trained on ImageNet-1K (Deng et al., 2009) which has a custom non-commercial license (see https://www.image-net.org/download.php). We also evaluate ADE20K, VOC2012, and BMC. For the licenses, see https://ade20k.csail.mit.edu/terms/index.html, http://host.robots.ox.ac.uk/pascal/VOC/voc2012/, and https://www.cancerimagingarchive.net/data-usage-policies-and-restrictions/, respectively.

**Images.** All the images in this paper are original and taken by the authors. Similarly, illustrations are created by the authors. The assets allow for non-commercial use and redistribution with proper attribution (CC BY-NC).

**Code.** The efficient octic layers are our original work and will be licensed under Apache License 2.0 following the code release. The training pipelines are from DeiT III (Touvron et al., 2022) and DINOv2 (Oquab et al., 2024), which are also under Apache License 2.0. For details about the Apache License 2.0 we refer to https://www.apache.org/licenses/LICENSE-2.0.

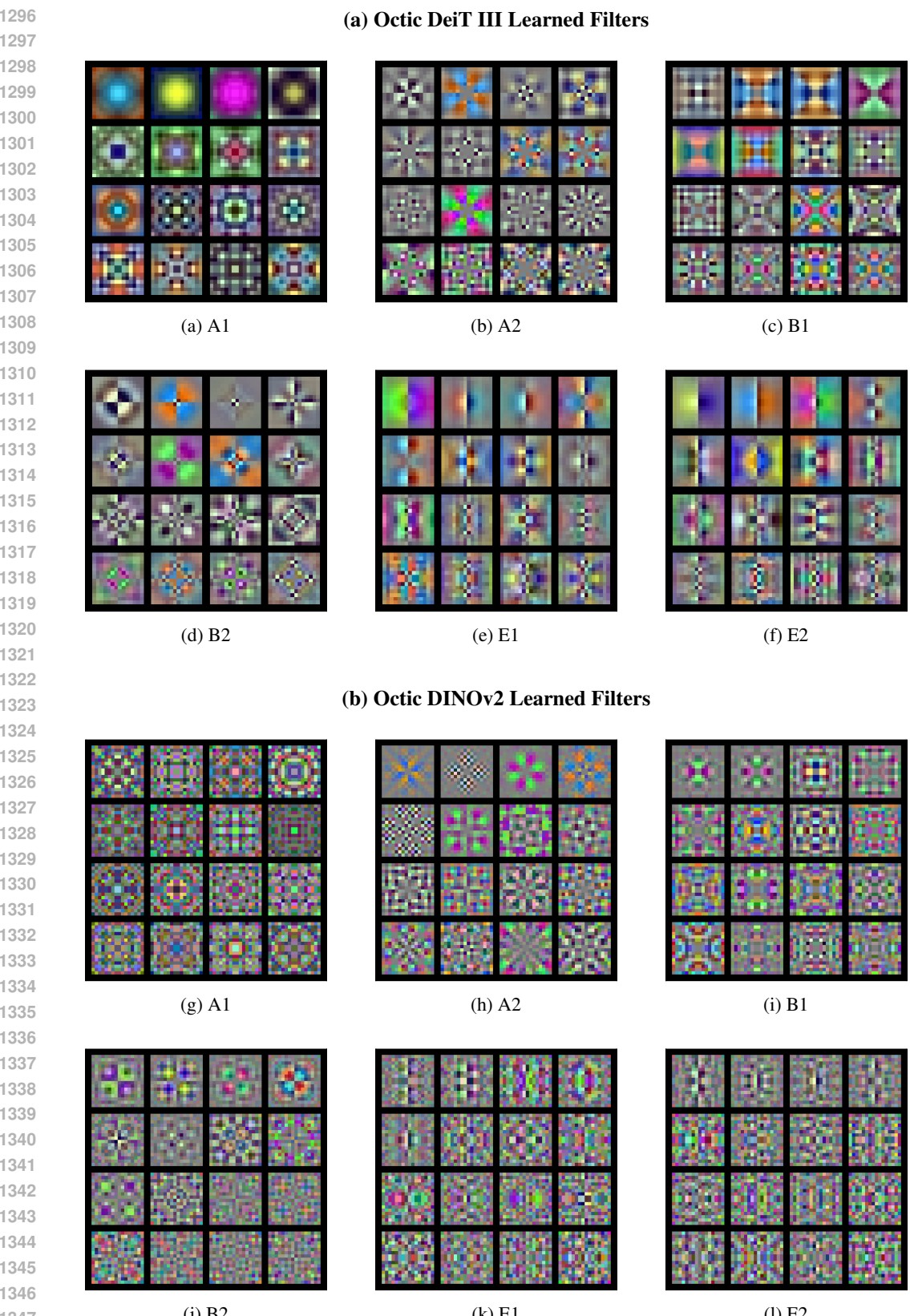

Figure 6: **Comparison of learned patch embedding filters.** (a) DeiT III. (b) DINOv2. Each figure shows the top-16 principal components of the octic PatchEmbed filter for a specific feature type.

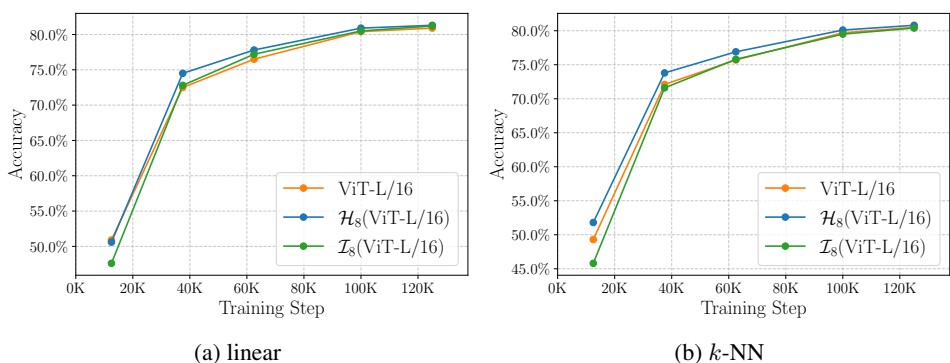

(a) linear

(b) $k$-NN

Figure 7: **DINOv2 training progression**. Classification accuracy development during 125K training steps for linear probe and $k$-NN on frozen features for ViT-L sized models.

