# OpenReview forum: "Octic Vision Transformers: Quicker ViTs Through Equivariance"
_ICLR.cc/2026/Conference — Submitted to ICLR 2026_

### Official Review · Reviewer_PPBJ · 2025-10-23

**Soundness:** 2
**Presentation:** 3
**Contribution:** 2
**Rating:** 4
**Confidence:** 4

**Summary:**

This paper introduces Octic Vision Transformers (octic ViTs) that leverage D8 group equivariance (90-degree rotations and reflections) to achieve computational efficiency gains in Vision Transformers. The key innovation is implementing equivariant linear layers in the Fourier domain of D8, achieving 5.33× FLOPs reduction and 8× memory reduction per feature dimension compared to standard linear layers. The authors propose two ViT families: I8 (fully equivariant) and H8 (breaking equivariance in later layers), demonstrating ~40% FLOP savings on ViT-H while maintaining accuracy on ImageNet-1K using both supervised (DeiT-III) and self-supervised (DINOv2) training recipes.

**Strengths:**

1. **Solid theoretical foundation**: The application of group representation theory to ViTs is mathematically rigorous, with clear exposition of the D8 group structure and isotypical decomposition.

2. **Comprehensive experimental validation**:
   - Evaluation on both supervised (DeiT-III) and self-supervised (DINOv2) training
   - Extensive ablations: invariantization methods (6 variants), number of octic blocks, effect of breaking equivariance
   - Additional evaluations on white blood cell classification (DinoBloom)

3. **Practical implementation details**: Custom Triton kernels for fused operations, efficient FFT implementation, and clear computational complexity analysis.

4. **Reproducibility**: Detailed hyperparameters, compute resources documented (Table 10), commitment to release code and weights.

5. **Honest reporting**: Authors acknowledge limitations regarding throughput not matching FLOPs savings.

**Weaknesses:**

### Major Issues:

1. **Theory-practice gap is severe**:
   - The 5.33× FLOPs reduction only translates to 1.47-2.37× throughput gains
   - Appendix B reveals octic layers only have better arithmetic intensity at C > 3200, well beyond experimental scales
   - This fundamentally undermines the efficiency claims and suggests limited practical applicability

2. **No accuracy improvements**:
   - All octic models only *match* baselines, never exceed them
   - For a top-tier conference with <10% acceptance, matching performance with some efficiency gains is insufficient
   - The accuracy-efficiency trade-off is not favorable enough to warrant publication at this venue

3. **Limited experimental scope**:
   - Only ImageNet-1K pretraining (1.2M images)
   - No evaluation on modern large-scale datasets (ImageNet-21K, JFT, etc.)
   - No real-world application demonstrations beyond standard benchmarks
   - Missing comparisons with other efficiency methods (pruning, quantization, knowledge distillation)

4. **Incremental novelty**:
   - Direct extension of Bökman et al. (2025) from D2 to D8
   - The key technical contribution (Fourier domain implementation) is relatively straightforward application of known theory
   - No fundamental architectural innovations

5. **Scalability concerns**:
   - Authors acknowledge they "do not scale the size beyond ViT-H" (Limitations, Section 5)
   - No experiments on ViT-G, ViT-e, or ViT-22B despite including them in Table 1
   - Unclear if benefits hold at truly large scales where efficiency matters most

### Minor Issues:

6. **Missing baselines**: No comparison with other structured/efficient transformers (e.g., local attention, sparse attention, linear attention variants)

7. **Evaluation limitations**:
   - OOD rotation evaluation is limited—only random 90° rotations, not continuous angles
   - Segmentation results show I8 models sometimes underperform (ADE20K: 33.9 vs 34.7 mIoU)

8. **Implementation dependency**: Requires custom Triton kernels, limiting accessibility and reproducibility for the broader community

**Questions:**

1. **Critical**: Can you explain why the arithmetic intensity analysis (Appendix B) shows octic layers only become advantageous at C ≈ 3200, yet you claim efficiency gains at C=1280? This appears contradictory.

2. **Scalability**: Table 1 projects improvements for ViT-G through ViT-22B. Why weren't these models actually trained? What evidence suggests the benefits will hold at larger scales?

3. **Comparison with other efficiency methods**: How do your models compare with:
   - Standard pruning/quantization techniques?
   - Knowledge distillation from larger models?
   - Other efficient ViT architectures (e.g., DeiT-distilled, EfficientViT)?

4. **Throughput analysis**: Can you provide a detailed breakdown of where the gap between FLOPs savings (5.33×) and throughput gains (1.47×) comes from? Is this fundamental or implementation-limited?

5. **Accuracy ceiling**: Is there a theoretical reason octic models cannot exceed baseline accuracy? Have you tried longer training or different optimization strategies?

6. **Real-world applications**: Can you demonstrate benefits on practical downstream tasks (object detection, instance segmentation, video understanding)?

7. **Continuous rotations**: How do I8 models perform under continuous rotation angles, not just 90° increments?

---

> ### Author Response · Authors · 2025-11-19
> **Response to Reviewer PPBJ (Weaknesses)**
>
> We thank the reviewer for the review and aim to answer their concerns and questions below. In this comment we address the outlined weakneses and in the following comment we answer the posed questions.
>
> ## Weaknesses
>
> > Theory-practice gap is severe
>
> See answers to specific questions under *Questions* below.
>
> > No accuracy improvements
>
> We motivate our work not by accuracy improvements but by significant computational savings. It is worth noting that we maintain or slightly improve accuracy while providing these efficiency gains.
>
> > For a top-tier conference with <10% acceptance, matching performance with some efficiency gains is insufficient
>
> This blanket statement is not part of the ICLR guidelines as far as we are aware. Furthermore, crucially, this reduction of the work leaves out the conceptual addition to the debate between hard-coding inductive biases and learning them from data. (As an aside, the number mentioned by the reviewer is incorrect as the acceptance rate at ICLR last year was 32%, see https://media.iclr.cc/Conferences/ICLR2025/ICLR2025_Fact_Sheet.pdf )
>
> > Limited experimental scope
>
> See answers to specific questions under *Questions* below.
>
> > No evaluation on modern large-scale datasets (ImageNet-21K, JFT, etc.)
>
> We use the standard large-scale dataset ImageNet-1K which we believe is enough to demonstrate the benefits of our method. Further, note that JFT is an internal dataset at Google so it is not possible for us (or indeed anyone in the community outside of Google) to evaluate on it. We refer to the last paragraph in the response to Reviewer b4ur for further discussion.
>
> > Incremental novelty
>
> We will happily concede that all group theory required for this paper has been known for over a hundred years. We do not see this as a “Major issue”, but rather a strength in that one can learn the required theory in a few lectures/chapters of a standard textbook in representation theory. The contribution in this work is in the design and evaluation of a ViT that is simultaneously equivariant and faster than a standard ViT, the currently dominant model in computer vision.
>
> > Scalability concerns
>
> See answers to specific questions under *Questions* below.
>
> > No comparison with other structured/efficient transformers (e.g., local attention, sparse attention, linear attention variants)
>
> These methods are compatible with our networks. See further the answer to a similar question below.
>
> > Requires custom Triton kernels, limiting accessibility and reproducibility for the broader community
>
> Triton itself is open source, fully integrated with PyTorch, and we will release the Triton kernels as open source. The custom kernels are therefore both accessible and reproducible.
>
> **Response continues in the following comment**

---

> > ### Author Response · Authors · 2025-11-19
> > **Response to Reviewer PPBJ (Questions)**
> >
> > ## Questions
> >
> > > Can you explain why the arithmetic intensity analysis (Appendix B) shows octic layers only become advantageous at C ≈ 3200, yet you claim efficiency gains at C=1280? This appears contradictory.
> >
> > This is in fact **not** contradictory. What matters for throughput is not only arithmetic intensity but also (for instance) the actual amount of FLOPs and the implementation. This is the reason why throughput needs to be measured rather than just inferred via arithmetic intensity or number of FLOPs. For instance a pointwise ReLU layer has lower arithmetic intensity than a fully connected Linear layer, but the ReLU layer is faster anyway due to a lower amount of FLOPs. Similarly, our linear layers require substantially (5.33$\times$) fewer FLOPs. Furthermore, the arithmetic intensity will not impact the throughput as long as the operation is compute bound, see e.g. https://jax-ml.github.io/scaling-book/ referenced in the manuscript for further discussion.
> >
> > > Table 1 projects improvements for ViT-G through ViT-22B. Why weren't these models actually trained?
> >
> > The numbers in Table 1 are not projections, but real measurements of throughput, since we can measure throughput using untrained networks (we measure inference throughput on an A100-80GB). The reason for not training them is that we do not have the same computational resources as the authors (a large group at Google) who proposed ViT-22B.
> >
> > > What evidence suggests the benefits will hold at larger scales?
> >
> > We demonstrate that the accuracy is maintained or improved at, in an academic context, large scale (ViT-H DINOv2) and show that the computational benefits only increase with scale (see e.g. Table 1 and Figure 4 (a)).
> >
> > > Comparison with other efficiency methods
> >
> > The proposed architecture is fully compatible with pruning, quantization and distillation. There is nothing prohibiting the use of those methods when using an equivariant model, ours in particular.
> >
> > EfficientViT proposes putting more compute into FFN parts of ViTs and less into MHSA. This is also fully compatible with our approach, in fact our approach focuses on making the FFN parts faster so it would be particularly helpful for EfficientViT.
> >
> > We choose to focus on standard ViTs in the paper to not confuse the main message, which is that ViTs gain efficiency from making them equivariant.
> >
> > > Can you provide a detailed breakdown of where the gap between FLOPs savings (5.33×) and throughput gains (1.47×) comes from? Is this fundamental or implementation-limited?
> >
> > The gap stems mainly from a throughput gap in the linear layers. This can likely be improved with better implementations and is also highly hardware dependent. Please see the answer to a similar question from EBbD for a more detailed answer.
> >
> > > Is there a theoretical reason octic models cannot exceed baseline accuracy? Have you tried longer training or different optimization strategies?
> >
> > There is no reason, in fact often the octic models do obtain slightly higher accuracy in our experiments. We have not tried other training recipes, longer training would very likely help. In response to a question from reviewer EBbD we tuned the weight decay parameter for a $\mathcal{H}_8(\text{ViT-B})$finding that it led to slightly improved performance. Please see the answer to EBbD for more details. As mentioned to reviewer EBbD, we find our current setup quite compelling in showing that simply switching the architecture to octic ViTs, without hyperparameter tuning, accuracy can be maintained while significantly reducing FLOPs.
> >
> > > Can you demonstrate benefits on practical downstream tasks (object detection, instance segmentation, video understanding)?
> >
> > We include results on segmentation (ADE-20K and VOC2012) in Table 3 as well as  hematology finetuning (DinoBloom) in Table 9 in the appendix.
> >
> > > How do I8 models perform under continuous rotation angles, not just 90° increments?
> >
> > We thank the reviewer for the question and in response we performed an experimental investigation. We randomly rotate the evaluation set by increments of 45° and find that the performance of the $\mathcal{I}_8$ networks slightly degrades but remains significantly more robust than baseline networks. For example, $\mathcal{I}_8(\text{ViT-H})$ achieves a classification accuracy of 79.3% (vs. 84.7% for 90° rotations) whereas ViT-H achieves 71.5% (vs. 85.0% for upright images). Note, we slightly crop the images to maintain 224x224 resolution after rotations.

---

### Official Review · Reviewer_b4ur · 2025-10-29

**Soundness:** 2
**Presentation:** 2
**Contribution:** 3
**Rating:** 4
**Confidence:** 4

**Summary:**

The paper proposes a new, efficient D8 equivariant vision transformer. The authors report extensive experiments which supports their method.

**Strengths:**

The experiments show that the method has good efficiency, though not so comprehensive, which will be discussed below.

The proposed method is interesting. Applying octic group to ViT is reasonable and looks have wide application domain.

**Weaknesses:**

My first concern is that the mechanism is not well motivated. I would suggest the authors adding in more explaination why the proposed method has better efficiency in the introduction and method sections. Currently, the authors justify the method that leveraging a larger group can yield "faster, stronger, and more compact models" without further establishment of this claim. I agree this could be right, but this leads to two questions: (1) Why not use a group that is even larger? (2) Would it be possible to quantitatively link the size of group and efficiency?

Ideally, theoretical analysis would be beneficial. For example, it would be interesting to see convergence or generalisation analysis linking the method with octic group.

Second, the experiments can be improved. The authors only use ImageNet-1k for classification, which is not sufficient. The compared methods are only Touvron et al. (2022) and Bökman et al. (2025). I would suggest the authors giving more results.

**Questions:**

Please see above.

---

> ### Author Response · Authors · 2025-11-19
> **Response to Reviewer b4ur**
>
> We thank the reviewer for the review and aim to address their concerns below.
>
> > My first concern is that the mechanism is not well motivated. I would suggest the authors adding in more explaination why the proposed method has better efficiency in the introduction and method sections. Currently, the authors justify the method that leveraging a larger group can yield "faster, stronger, and more compact models" without further establishment of this claim. I agree this could be right, but this leads to two questions: (1) Why not use a group that is even larger? (2) Would it be possible to quantitatively link the size of group and efficiency?
>
> The efficiency gain stems from making the linear layers equivariant and parameterizing them in the Fourier domain, as laid out in Section 3.3.2 and Figure 2. We will reformulate the second paragraph of the Introduction to be clearer and appreciate the reviewer raising this concern. Below we address the reviewer’s two questions:
>
> (1) Using larger groups is possible, but there is a tradeoff in the fact that using larger groups leads to fewer trainable parameters and more complicated implementations. We believe that studying one group in depth makes sense for this paper.
>
> (2) The direct quantitative link is that the more irreps a group has, the faster the linear layers will become. For common groups acting on images, the irreps are well-known from mathematics. For instance, dihedral groups of order $4n$ (i.e. $180/n$ degree rotations and reflections) have 4 one-dimensional irreps and $n-1$ two-dimensional irreps. The regular representation splits into the four 1D irreps and two copies each of the 2D irreps yielding a total of $4+(n-1)\cdot2\cdot2=4n$ dimensions. As the irreps of dihedral groups are the same over complex and real numbers (which is not the case for all groups), we can apply Schur’s lemma over the complex numbers, simplifying the theory compared to the real valued case. Schur’s lemma says that we get 4 blocks of size 1 and $n-1$ twice repeated blocks of size 2 in linear layers from the regular representation to itself. In Figure 2 of the submission, this is visualized with $n=2$ to get the octic group. The number of multiplications in an equivariant linear layer will be $4+ (n-1)\cdot2^2\cdot2 =4(2n-1)$ and the compute savings relative to an ordinary linear layer is $\frac{(4n)^2}{4(2n-1)}=\frac{4n^2}{2n-1}$ times. For the case $n=2$ as in the paper we obtain the familiar saving of $16/3$, i.e. 5.33$\times$. (Note that we have left out the number of channels in the analysis above, but using $4n\cdot c$ channels with $c>1$ simply adds a factor $c^2$ to both the equivariant and ordinary linear layers so we obtain the same result.)
>
> We believe that including the above discussion makes for a nice addition to the paper and thank the reviewer for the suggestion.
>
> > Ideally, theoretical analysis would be beneficial. For example, it would be interesting to see convergence or generalisation analysis linking the method with octic group.
>
> Is there convergence or generalisation analysis for standard ViTs of the type the reviewer has in mind? Then we could potentially relate that analysis to the octic equivariant versions.
> Please note that we present a thorough theoretical analysis regarding computational complexity and convergence of FLOPs and throughput savings as the width of the network grows (see e.g. Table 1 and 4 and Figure 4 (a)) .
>
> > Second, the experiments can be improved. The authors only use ImageNet-1k for classification, which is not sufficient. The compared methods are only Touvron et al. (2022) and Bökman et al. (2025). I would suggest the authors giving more results.
>
> We respectfully disagree with the assessment. In addition to supervised training with DeiT III as in Touvron et al. and Bökman et al., we also include self-supervised training with DINOv2. Further we include DinoBloom in the appendix. Focusing on classification on ImageNet is standard and follows prior work on ViTs, but note that the experiments using DINOv2 also evaluate on segmentation on both ADE20K and VOC2012. We provide much more extensive experiments than commonly found in the literature on equivariant methods for image data, where experiments are typically restricted to small datasets such as CIFAR10 or MNIST. Note, it is also common in prior work to evaluate on rotated versions of said datasets. In contrast, we replicate state-of-the-art evaluation pipelines for standard ViTs.

---

### Official Review · Reviewer_EBbD · 2025-10-31

**Soundness:** 3
**Presentation:** 3
**Contribution:** 3
**Rating:** 6
**Confidence:** 3

**Summary:**

This paper introduces Octic ViT, a Vision Transformer leveraging octic group equivariance. The core innovation is an efficient octic-equivariant linear layer that operates in the Fourier domain, reducing the linear layer's FLOPs by 5.33x and parameters by 8x. Experiments (DeiT-III, DINOv2) show that Octic ViTs match or exceed the accuracy of ViT-H/L baselines while saving ~40% of total FLOPs.

**Strengths:**

- Significant Compute Efficiency: Drastically reduces FLOPs for SOTA ViTs without sacrificing accuracy.

- Efficient Equivariance: Unlike prior work where equivariance adds overhead, this paper uses Fourier domain sparsity to accelerate ViTs.

- SOTA Validation: Validated at scale on SOTA training recipes, proving practical utility.

- Architectural Insights: The ablation between hybrid and fully invariant models provides valuable design insight.

**Weaknesses:**

- FLOPs vs. Throughput Mismatch: The large FLOPs reduction does not fully translate to throughput/speedup (only 1.47x).

- Implementation Complexity: The method is complex, requiring knowledge of group representation theory, Fourier transforms, and custom Triton kernels.

- Non-linearity Overhead: GELU activations must be applied in the spatial domain ($\rho_{reg}$), requiring costly round-trips via Fourier transforms.

**Questions:**

- Throughput Bottleneck: What is the primary bottleneck causing the gap between FLOPs savings (4.6x) and throughput gain (1.5x) for ViT-H? Is it the custom Triton kernel or the Fourier transforms?

- Hybrid vs. Invariant: Why do the hybrid $\mathcal{H}_{8}$ models almost always achieve slightly better accuracy than the fully invariant $\mathcal{I}_{8}$ models? Does this imply ImageNet requires non-symmetric features?

- Hyperparameter Tuning: You did not retune hyperparameters despite an 8x parameter reduction in the linear layers. Does this suggest the model is robust, or could accuracy be further improved by tuning regularization (e.g., weight decay)?

---

> ### Author Response · Authors · 2025-11-19
> **Response to Reviewer EBbD (Throughput Bottleneck)**
>
> We thank the reviewer for their review and answer the questions below. We have divided the response into two comments. The first one addresses the first question (about throughput) and the second the rest.
>
> ## Throughput Bottleneck
>
> > Throughput Bottleneck: What is the primary bottleneck causing the gap between FLOPs savings (4.6x) and throughput gain (1.5x) for ViT-H? Is it the custom Triton kernel or the Fourier transforms?
>
> As seen in Table 4 in the appendix, the custom Triton kernel for iFFT->GELU->FFT is almost as fast as standard GELU. The main bottleneck is actually the matrix multiplication in the linear layers.
>
> At the dimensionality of the linear layers in ViT-H, the octic layers from C dimensions to 4C dimensions are just barely compute bound using fp16 matrix multiplication on A100 tensor cores. This can be seen by computing the arithmetic intensity as presented in Appendix B. With $P=2$, batch size $B=64\cdot196$, $C=1280$ and $F=4C$, we obtain an arithmetic intensity (AI) of 190 which is quite close to the ridge point of 156 FLOPs/byte at which computations become theoretically compute bound on A100 80GB gpus (see https://modal.com/gpu-glossary/perf/arithmetic-intensity). In contrast, ordinary linear layers have an AI of 950 and hence much more leeway in implementation. Note that if we would use fp32 precision ($P=4$), then octic linears have an AI of 95 and standard linears 475 which are both well above the ridge point for fp32 at 10 FLOPs/byte (much lower since there are no tensor cores for fp32). On GPUs without tensor cores, octic linears are also much more compute bound.
>
> We have conducted a throughput benchmark on just the linear layers to make this point clearer and will update the manuscript to include it. We compare standard linear layers to octic linear layers and a simple blockdiagonal linear layer with 8 equal size blocks, which uses 8 times fewer FLOPs than a standard linear layer. We report the performance of each implementation in TFLOP/s, which should ideally be close to 312 for fp16 and 19.5 for fp32 (the maximum achievable on A100 80GB gpus). Of course even with suboptimal performance, the layers with fewer FLOPs can be faster than the baseline which is what we observe.
>
> | Layer                  | dtype       | Channels         | AI (FLOPs/byte)   | Time (ms) | Performance (TFLOP/s) | Relative speedup   |
> |------------------------|-------------|------------------|-------------------|-----------|-----------------------|--------------------|
> | nn.Linear              | fp16        | 1024 -> 4096     | 770               | 0.47      | 230                   | 1.0                |
> | Octic                  | fp16        | 1024 -> 4096     | 150               | 0.19      | 110                   | 2.6                |
> | Blockdiag              | fp16        | 1024 -> 4096     | 100               | 0.13      | 100                   | 3.6                |
> | nn.Linear              | fp16        | 8192 -> 32768    | 4300              | 24        | 280                   | 1.0                |
> | Octic                  | fp16        | 8192 -> 32768    | 1200              | 5.3       | 240                   | 4.6                |
> | Blockdiag              | fp16        | 8192 -> 32768    | 770               | 3.4       | 250                   | 7.2                |
> | nn.Linear              | fp32        | 1024 -> 4096     | 380               | 5.6       | 19                    | 1.0                |
> | Octic                  | fp32        | 1024 -> 4096     | 80                | 1.2       | 17                    | 4.8                |
> | Blockdiag              | fp32        | 1024 -> 4096     | 50                | 0.76      | 17                    | 7.4                |
> | nn.Linear              | fp32        | 8192 -> 32768    | 2200              | 360       | 19                    | 1.0                |
> | Octic                  | fp32        | 8192 -> 32768    | 580               | 67        | 19                    | 5.33               |
> | Blockdiag              | fp32        | 8192 -> 32768    | 380               | 45        | 19                    | 7.95               |
>
> Note that matmul-kernels are heavily optimized for the standard nn.Linear case so it should be possible to close the gap in performance by working on specializing to block-diagonal linear layers of the type used in our equivariant networks (especially since the layers are theoretically compute bound for C=1280 even in fp16).
>
> We thank the reviewer for raising an interesting question and believe the above discussion to be a valuable addition to the paper.

---

> ### Author Response · Authors · 2025-11-19
> **Response to Reviewer EBbD (Remaining Questions)**
>
> ## Remaining Questions
>
> > Hybrid vs. Invariant: Why do the hybrid H_8 almost always achieve slightly better accuracy than the fully invariant I_8 models? Does this imply ImageNet requires non-symmetric features?
>
> The $\mathcal{H}_8$ models are more flexible. Some semantic features useful for ImageNet are not rotation invariant, since images in the dataset are almost always upright. For instance, there is no need for the model to be able to recognize an upside down elephant for good performance on ImageNet. Since the network has finite capacity, it can lead to better performance on ImageNet to recognize many classes in upright orientation than to recognize fewer classes in all orientations.
>
> > Hyperparameter Tuning: You did not retune hyperparameters despite an 8x parameter reduction in the linear layers. Does this suggest the model is robust, or could accuracy be further improved by tuning regularization (e.g., weight decay)?
>
> We thank the reviewer for the question and in response we conducted a small scale experiment on DeiT III training of ViT-B. We found that an approximate 0.2 ppts. accuracy increase could be achieved for $\mathcal{H}_8\text{(ViT-B)}$ by tuning the weight decay. This indicates that our results could slightly benefit from hyperparameter tuning. However, we find our current setup quite compelling in showing that simply switching the architecture to octic ViTs, without hyperparameter tuning, can maintain accuracy while significantly reducing FLOPs.

---

### Official Review · Reviewer_5pGM · 2025-11-02

**Soundness:** 3
**Presentation:** 2
**Contribution:** 2
**Rating:** 2
**Confidence:** 3

**Summary:**

This paper proposes using octic group equivariant ViTs for better performance in downstream tasks. The authors use custom Triton kernels to speed up training equivariant ViTs and introduce different variants where $I_8$ is exactly invariant in the later layers and $H_8$ break equivariance in the later layers. Experiments on ImageNet-1k show that $H_8$ can outperform standard ViTs.

**Strengths:**

- Good comparison of end-to-end equivariant ($D_8$), late invariant ($I_8$), late non-equivariant ($H_8$) and their performance.
- Extensive experiments on ImageNet classification and an SSL task show that equivariance can help.

**Weaknesses:**

A critical weakness of this paper in my opinion is the limited novelty. Equivariant ViTs have been proposed previously [1, 2, 3] and octic ViTs are a special case. It is also well known that weight sharing/weight tying reduces the number of FLOPs as a direct consequence of the reduction of the number of parameters [4, 5]. Furthermore, several papers have shown that some form of symmetry breaking, especially in the later layers, can be beneficial for CNNs [4, 5, 6], albeit not for ViTs. Thus the novelty is minimal and thus I feel that the main contribution of this paper lies on the fact that the authors provide a custom Triton kernel to speed up wall clock time and do extensive FLOPs analysis on the variants.

Another critical weakness is that the argument that equivariance is beneficial in ViTs goes against the results. The variant $H_8$, which uses non-equivariant layers in the later blocks performs the best, beating regular ViTs, but the exactly equivariant ViT $D_8$ or even the invariant ViT $I_8$ perform worse than a standard ViT. There should be a nuanced discussion about why equivariance, specifically in the earlier layers, helps over the blanket statement that equivariance is beneficial and should be used more if it could be sped up.

Lastly, the FLOPs seems to decrease but the wall clock time is higher for all variants. Table 10 in the Appendix shows that the ViT-L/16 baseline takes 24 hours, compared to 39 for $H_8$ or 37 for $I_8$. This contradicts the statement that the proposed methods are indeed faster. The reason why equivariant ViTs require lower number of FLOPs is due to the weight tying/sharing reducing the number of parameters, but the wall clock time is often a lot slower than regular ViTs because the representation must be transformed to the spatial domain ($\rho_{iso}$ to $\rho_{reg}$) in order to apply the nonlinearities pointwise, which are then mapped back to $\rho_{iso}$ after the nonlinearity. This is true for equivariant steerable convolutions as well.

One way I can see to improve the paper in the future would be to focus on the symmetry breaking aspect, and deeply investigate why breaking equivariance in later layers, for ViTs, helps (perhaps by looking at the intermediate representations).

References
1. Romero, D. W., & Cordonnier, J. B. (2020). Group equivariant stand-alone self-attention for vision. arXiv preprint arXiv:2010.00977.
2. Xu, R., Yang, K., Liu, K., & He, F. (2023, July). $ E (2) $-Equivariant Vision Transformer. In Uncertainty in Artificial Intelligence (pp. 2356-2366). PMLR.
3. Klee, D., Park, J. Y., Platt, R., & Walters, R. A Comparison of Equivariant Vision Models with ImageNet Pre-training. In NeurIPS 2023 Workshop on Symmetry and Geometry in Neural Representations.
4. Weiler, M., & Cesa, G. (2019). General e (2)-equivariant steerable cnns. Advances in neural information processing systems, 32.
5. Kondor, R., & Trivedi, S. (2018, July). On the generalization of equivariance and convolution in neural networks to the action of compact groups. In International conference on machine learning (pp. 2747-2755). PMLR.
6. Vadgama, S., Islam, M. M., Buracas, D., Shewmake, C., Moskalev, A., & Bekkers, E. (2025). Probing Equivariance and Symmetry Breaking in Convolutional Networks. arXiv preprint arXiv:2501.01999.

**Questions:**

I'll write some minor comments here.
- The octic group is more commonly denoted as $D_4$ rather than $D_8$. I would recommend changing to the more common notation.
- The preliminaries seem a touch too long, given that these are known results. Some of the preliminaries also don't seem to be necessary to understand the method.

---

> ### Author Response · Authors · 2025-11-19
> **Response to Reviewer 5pGM**
>
> We thank the reviewer for the thorough review and aim to rebut their concerns in the following.
>
> We first wish to clarify a misunderstanding of the reviewer regarding wall clock times.
>
> > Lastly, the FLOPs seems to decrease but the wall clock time is higher for all variants. Table 10 in the Appendix shows [...]
>
> Table 10 should not be interpreted as a fair comparison of training throughput, but rather an accurate compute reporting for the resources used for this research project. Below, we have summarized the training (forward+backward) speeds (throughput in images/second, higher is better) for all our model types in a fair comparison. As illustrated, the speed gains at inference presented in Table 2 translate into training speed gains.
>
> | Model                         | DeiT III | DINOv2 |
> |-------------------------------|----------|--------|
> | $\text{ViT-H}$                | 180      | 94     |
> | $\mathcal{I}_8(\text{ViT-H})$ | 207      | 114    |
> | $\mathcal{H}_8(\text{ViT-H})$ | 209      | 115    |
> | $\mathrm{D}_8(\text{ViT-H})$  | 257      | 142    |
> | $\text{ViT-L}$                | 454      | 184    |
> | $\mathcal{I}_8(\text{ViT-L})$ | 510      | 210    |
> | $\mathcal{H}_8(\text{ViT-L})$ | 512      | 214    |
> | $\mathrm{D}_8(\text{ViT-L})$  | 564      | 232    |
>
> Even further gains relative to the baseline are obtained if the training is done in fp32 precision. E.g., $\mathcal{H}_8(\text{ViT-L})$ speedup increases from approximately 14% to approximately 50%.
>
> The reason for the discrepancy in Table 10 is unrelated to the efficiency of our equivariant layers. We mention the reason in the caption, “*Efficiency gains were made progressively and thus some experiments took longer than necessary. Similarly, our training pipeline for DINOv2 is not fully optimized for the octic layers*” and reference it on line 1075 “*[...] the octic layers do not leverage NestedTensorBlock and training speedups associated with xFormers. This choice does not impact speed on downstream tasks but slightly decreases pre-training speed*”. The reason is thus simply that, at the time of training DINOv2, we did not implement specific training speedups, because they do not matter for inference throughput. We apologize for the vague formulation in the caption of Table 10 and will address this in the manuscript. Furthermore, the training times for $\mathrm{D}_8$ in Table 10 do not accurately reflect the current training speed. Those networks were trained before we optimized the Tensor shapes and the non-linearity kernel. The $\mathcal{H}_8$ and $\mathcal{I}_8$ networks were trained with the optimized implementation.
>
> Lastly, in addition to the training throughput table above, a fair comparison is also given in the inference throughput benchmarks, e.g. Tables 1 & 2. There the throughput benefit of our implementation is clear to see, which distinguishes our work from the prior literature since, as the reviewer writes:
>
> > the wall clock time is often a lot slower [for equivariant] than regular ViTs
>
> The reviewer also writes:
>
> > The reason why equivariant ViTs require lower number of FLOPs is due to the weight tying/sharing reducing the number of parameters, but the wall clock time is often a lot slower than regular ViTs because the representation must be transformed to the spatial domain ($\rho_{reg}$) in order to apply the nonlinearities pointwise, which are then mapped back to ($\rho_{iso}$) after the nonlinearity.
>
> In general, having fewer parameters does not automatically lead to fewer FLOPs. For instance in GCNNs, parameters are expanded out so that each parameter is used in more operations than in an ordinary CNN. Note that we do **not** expand the weights using “weight tying” since working in the Fourier domain enables us to just use the Fourier space weights without expanding them to the spatial domain.  The overhead for transforming back and forth to the Fourier domain around each GELU-nonlinearity is actually negligible in our implementation, refer to Table 4 in the appendix.
>
> **Next, we address the remaining concerns in the following comment.**

---

> ### Author Response · Authors · 2025-11-19
> **Continuation of response to Reviewer 5pGM**
>
> **Below we adress the remaining concerns.**
>
> > Equivariant ViTs have been proposed previously [1, 2, 3] and octic ViTs are a special case.
>
> While related, octic ViTs are not a special case of the ViTs in [1, 2]. As the reviewer correctly notes, those methods do not obtain computational benefits over standard ViTs: They implement attention over all image positions as well as all orientations in the symmetry group $G$ (after having lifted the input image to a signal over positions and orientations) which increases complexity.
>
> In contrast, we only use attention between image positions, just like ordinary ViTs, and instead have steerable features at each image position similar to steerable CNNs. This enables us to get computational benefits relative to standard ViTs in the linear layers while not altering the attention mechanism and hence avoiding additional costs for attention, unlike [1, 2].
>
> [3] does not include equivariant ViTs beyond this sentence as far as we can tell “*Currently, only ResNet-style architectures are implemented and trained; we plan to add equivariant vision transformers (introduced by Romero and Cordonnier (2020)), and are open to supporting other models.*”.
>
> > It is also well known that weight sharing/weight tying reduces the number of FLOPs as a direct consequence of the reduction of the number of parameters [4, 5].
>
> We believe the above statement reflects a misunderstanding. In [4], the number of FLOPs is kept the same as a CNN with the same expanded convolution filter sizes and thus the FLOPs are not reduced by making the CNN equivariant. The authors of [4] make a point out of the fact that the FLOPs are the same: “*The resulting kernels are then used in a standard convolution routine. [...] In evaluation mode the parameters are not updated such that the kernel needs to be expanded only once and can then be reused. E(2)-steerable CNNs therefore have no computational overhead in comparison to conventional CNNs at test time.*” Thus, FLOPs are not reduced.
>
> In [5], no networks are implemented and we could not find any discussion of computational complexity.
>
> In our networks, equivariant layers are made strictly more efficient compared to non-equivariant ones. Neither [4] nor [5] presents any such computational benefits.
>
> > Furthermore, several papers have shown that some form of symmetry breaking, especially in the later layers, can be beneficial for CNNs [4, 5, 6], albeit not for ViTs.
>
> We cite [4] when discussing the symmetry breaking on line 464. We do not claim to be the first to observe the benefits.
>
> > There should be a nuanced discussion about why equivariance, specifically in the earlier layers, helps over the blanket statement that equivariance is beneficial and should be used more if it could be sped up.
>
> We agree that this discussion should be improved and thank the reviewer for the suggestion. In particular, as the reviewer has mentioned, our findings for breaking equivariance corroborate earlier findings from other works and we will edit the section around line 464 to make this clearer and also cite [6]. However, the main point in this work is that making a ViT equivariant can make it faster if done in the way proposed in the paper. Hence even if equivariance “only” maintains performance, it can be beneficial for the sake of computational efficiency.
>
> > The octic group is more commonly denoted as $D_4$ rather than $D_8$. I would recommend changing to the more common notation.
>
> This is a matter of convention. For instance wikipedia uses $D_4$ while nLab and group-props use $D_8$. We prefer $D_8$ since it illustrates that the group has order 8 and is consistent with our naming octic ViTs, but are open to change if the reviewer believes this is critical.
>
> > The preliminaries seem a touch too long, given that these are known results. Some of the preliminaries also don't seem to be necessary to understand the method.
>
> We are also open to moving some preliminaries to the appendix and adding the ablations on different types of invariantizations to the main paper instead. However, we do think that the preliminaries are required to understand the method and in particular the reason why our networks are fast relative to non-equivariant networks. Readers experienced in group theory can easily skip ahead, but many readers interested in ViTs are not familiar with the Fourier transform on finite groups and will hopefully benefit from the preliminaries.

---

### Author Response · Authors · 2025-11-19

We thank the reviewers for their time, effort and constructive feedback. We have answered each reviewer in detail and hope to have clarified some misunderstandings regarding the method. Further, we have added experiments and analyses in response to each reviewer which we believe strengthen the paper. Of interest to all reviewers is:

1. Detailed benchmarking of forward+backward as a response to 5pGM, which shows that the gains in inference speed also translate to reduced training time.
2. A study of throughput bottlenecks as a response to EBbD,  the results of which show that achieving sufficient arithmetic intensity in the linear layers is a key factor.
3. A theoretical discussion regarding scaling to larger dihedral groups in response to b4ur.
4. Benchmarking of performance under multiples of 45 degree rotations in response to PPBJ, showing that our model is significantly more robust, also in this setting, than standard ViTs.

We look forward to continuing the discussions with the reviewers and will update the manuscript later during the discussion phase, at which point we will write another global comment detailing the changes.

---

> ### Author Response · Authors · 2025-11-27
> **Updated manuscript**
>
> We have now updated the manuscript with the above analyses. The benchmarking of forward+backward is in Table 6. The benchmarking of throughput in linear layers is in Appendix B.1. The discussion regarding larger dihedral groups is in Appendix D. The evaluation under 45 degree rotations is in Table 12. Furthermore, we have edited the final paragraph of Section 4.3 to more clearly discuss prior work on symmetry breaking, on recommendation from Reviewer 5pGM and we have edited the second paragraph of the introduction to make it clear straight away where the compute savings come from, on recommendation from Reviewer b4ur.
>
> We thank the reviewers for their helpful feedback, leading to these improvements of the paper. We hope to hear back from the reviewers soon regarding remaining concerns, as we were quite early in posting responses to the reviews. If concerns have been addressed we humbly ask the reviewers to raise their scores.

---

### Meta-Review · Area_Chair_9vHj · 2025-12-26

**Summary:**

The paper proposes a $D_8$-equivariant ViT with steerable features, which leverages the sparsity of the equivariant linear layers to achieve gains in efficiency.

Initial reviews were mixed but leaning towards rejection. Reviewers valued the efficiency gains demonstrated through extensive experiments and ablations, the practical implementation and solid theoretical grounding. They raised concerns about novelty, and questioned the validity of the efficiency gains as well as the motivation.

Unfortunately the quality of the reviews was overall low with several factual errors and raising of irrelevant points. The rebuttal clarified many of those points. The major remaining concern is the one of novelty, since the paper can be seen as an extension of Bökman et al. (2025) to a larger group (as raised by PPBJ). The main motivation and ideas are the same, but the submission shows slightly better efficiency and/or accuracy, at the cost of increased complexity. I view this as a borderline submission so I'll side with most initial reviews and recommend rejection.

**Reviewer Concerns:**

5pGM concerns: The rebuttal clarified that it is not a special case of the mentioned references due to the steerability aspect and the efficiency gains enabled by it. The rebuttal also clarified some timing tables misunderstandings.

EBbD had minor concerns that were addressed.

b4ur concerns about motivation were partially addressed with a theoretical analysis but implementations and results are lacking.

PPBJ raised several irrelevant concerns but also raised the most important one, that the paper is a direct extension of Bökman et al. (2025) so novelty is limited. It was not addressed.

**Reviewer Scores:**

I believe 5pGM would increase their score because their main concerns seem to come from misunderstandings that were clarified -- perhaps to a 4 or 6.

b4ur main concern was only partially addressed and PPBJ most serious concern also was not addressed. I don't believe they would change their rejection recommendation.

---

### Decision · Program_Chairs · 2026-01-26

Reject